# Principal Spectral Regularization Makes Momentum Surpass Adam for LLM Training

## Abstract

Adam has been the most popular optimizer for training deep neural networks for nearly a decade. Recently, Muon, known for its momentum orthogonalization property, has emerged as a strong alternative to training large language models (LLMs). However, is orthogonalization over the whole learning space really necessary, especially given the high computational complexity of Newton-Schulz iteration in Muon? To the best of our knowledge, we are the first to report that Momentum with marginal spectral regularization on very few dimensions can surprisingly surpass Adam. In this work, we mainly made three contributions. First, from spectral visualizations of the LLM training dynamics and the optimization of the Styblinski-Tang function, we observe that the full orthogonalization of the matrix can be suboptimal in some cases. Second, we propose a novel principal spectral regularization (PSR) method that selectively penalizes only the dominant components with computational efficiency. Third, we show that the PSR approach enables SGD with momentum to surpass Adam in pretraining LLMs.

## 1 Introduction

Large language models (LLMs) have emerged as a breakthrough and state-of-the-art in various natural language processing (NLP) tasks in recent years. With pretraining on enormous text corpora, LLMs have demonstrated strong performance in handling responsibilities such as question answering (QA), code generation, and even research assistance, thereby enhancing industry production efficiency and contributing to broader social welfare (Kumar, 2024; Gao et al., 2025). However, pretraining LLMs from scratch remains highly resource-intensive for both academia and industry, requiring substantial infrastructure and specialized machine learning expertise, which motivates the search for new training techniques that prioritize efficiency.

Optimizers are at the core of training techniques as they dominate computational resources, while Adam (AdamW) (Kingma, 2014; Loshchilov & Hutter, 2017) is widely regarded as the "king" of optimizers (Bremen, 2020). By maintaining exponentially decaying moving averages of gradients and second-order momentum, Adam approximates per-parameter curvature with a diagonal approximation, offering stability and fast convergence across diverse neural architectures and tasks. Nevertheless, the memory overhead of storing and updating two moments with auxiliary tensors for all model parameters is significant in LLM pretraining when millions or billions of parameters are involved. Efforts to advance single-momentum optimizers to comparable performance with Adam and its variants remain an active area of research (Chen et al., 2023; Shu, 2023; Liang et al., 2024; Zhang et al., 2025; Glentis et al., 2025).

Recently, spectral methods for accelerating LLM pretraining have garnered increasing attention due to Muon's proven benefits in sample efficiency and reduced memory consumption compared to AdamW (Jordan et al., 2024; Liu et al., 2025a). Spectral regularization has been studied from multiple perspectives, including penalizing the spectral norm of parameters (Yoshida & Miyato, 2017; Miyato et al., 2018) or gradients (Lewandowski et al.). Another line of research is spectral preconditioning, explored in methods including preconditioned SGD (Li, 2017), Adafactor (Shazeer & Stern, 2018), Shampoo (Gupta et al., 2018), SOAP (Vyas et al., 2024), etc., for which matrix-shaped preconditioners are maintained instead of scalar preconditioners. Building on prior attempts, Muon employs the Newton-Schulz iteration to approximate the nearest semi-orthogonalization of momentum, aiming at achieving nearly uniform updates at all spectral directions. Some researchers

even consider Muon as a strong candidate for the next standard optimizer for LLM pretraining, according to its performance in practice (Team et al., 2025; Shah et al., 2025; Zeng et al., 2025).

In this work, by comparing Momentum, AdamW, and Muon from a spectral perspective, we notice that the spectral structure may play an essential role in LLM pretraining acceleration. We observe a *spiked-head-heavy-tail* structure in the original momentum covariance, while Muon has a very flat spectrum as it orthogonalized the momentum. However, in optimizing the Styblinski-Tang function, we discover that penalizing only a fraction of the dominant updates can yield better convergence compared to SGD-M, Adam, and Muon. This motivates a deeper investigation into how different spectral components and orthogonalizing them affect training efficiency, intending to interpolate between the computational trade-offs of full momentum orthogonalization. Building on our hypothesis, we propose a novel principal spectral regularization (PSR) method that selectively penalizes dominant directions using a simplified Lanczos bidiagonalization procedure and deflation. Our methodology is significantly more efficient than the Newton-Schulz iteration in theoretical complexity, which contributes to an in-depth understanding of optimization from a scalable perspective.

This work made three main contributions.

- Discovering the spectral visualization results of Momentum, Adam, and Muon, and the intuitive findings from optimizing the Styblinski–Tang function, we raise a key insight that orthogonalization over the whole space can be questionable for LLM training, due to the high computational complexity of Newton-Schulz iterations employed in Muon.
- Motivated by our insight, we propose a **principal spectral regularization (PSR)** method that selectively regularizes very few "spiked-head" components in the high-dimensional momentum, which is more efficient than Newton-Schulz theoretically in computational complexity and empirically in high-dimensional matrices.
- Our extensive experiments demonstrate that the proposed PSR method can help Momentum surpass AdamW in LLM pretraining over just a very few dimensions, which revealed the roles of different spectral components in the momentum spectra.

## 2 RELATED WORKS

This section reviews previous work on spectral methods in deep learning optimization and optimizers designed with efficiency in LLM pretraining with recent benchmarking experiments.

### 2.1 SPECTRAL METHODS IN DEEP LEARNING OPTIMIZATION

We consider and compare two types of spectral methods in optimizing deep learning models: spectral regularization and spectral preconditioning. Previous spectral regularization methods mostly estimate the spectral norm using the largest singular value of the weights or gradients, then add it as a penalty term to the loss to discourage low-entropy solutions (Yoshida & Miyato, 2017; Miyato et al., 2018; Lewandowski et al.). Spectral preconditioning optimizers typically leverage the spectral properties of some matrices associated with the model, usually the gradient/momentum or a Hessian approximation (e.g., the Fisher information (Martens & Grosse, 2015)), and rescale parameter updates along each spectral direction (Doikov et al., 2024). Early efforts include PSGD/Kron that approximate the inverse Hessian with Kronecker-factored preconditioners (Li, 2017; 2022), Adafactor with a rank-1 preconditioner for memory efficiency (Shazeer & Stern, 2018), Shampoo with a full-matrix preconditioner for each dimension of parameters (Gupta et al., 2018), and Sophia with an online estimate of the second-order preconditioner for scalability (Liu et al., 2023).

In 2023, Jordan et al. introduced Muon, an optimizer that performs efficient momentum orthogonalization by Newton-Schulz iterations to approximate the matrix sign function. This approach yields a semi-orthogonal momentum update matrix, effectively amplifying the 'rare directions', directions with small gradient components but potentially high importance for generalization (Jordan et al., 2024). Liu et al. further extended Muon by introducing a rescaling scheme that aligns its update RMS with AdamW. Muon and its variants have since been empirically validated in large-scale LLM pretraining, demonstrating improved sample efficiency (Liu et al., 2025a; Shah et al., 2025; Team et al., 2025; Zeng et al., 2025). These results have renewed interest in understanding spectral preconditioning and orthogonalization as fundamental tools for optimization in deep learning.

In contrast to previous spectral optimization methods, we notice a gap between regularization applied only to the top spectral direction or to the spectral norm and full-matrix preconditioning. Our proposed method bridges this gap by first identifying the principal spectral directions and then explicitly regularizing them, offering a scalable perspective and understanding of spectral methods.

## 2.2 OPTIMIZERS FOR LLM PRETRAINING

AdamW is the most widely used optimizer across deep learning architectures, including both pretraining and fine-tuning of decoder-only transformers (Zhao et al., 2024b). However, it stores both first- and second-moment estimates, effectively occupying twice the space of weights/gradients in GPU memory. This has motivated numerous efforts to reduce the memory footprint or improve sample efficiency, particularly for large-scale LLM training. Notable approaches include Adam-mini with block-wise learning rate schedules based on Hessian partitions (Zhang et al., 2024; Wang et al., 2025), Lion with momentum-sign updates (Chen et al., 2023) and FOCUS that enhance Signum with parameter attentions (Liu et al., 2025b), Cautious Adam/Lion with gradient-aligned selective updates (Liang et al., 2024), SWAN that enhances SGD with whitening and normalization (Ma et al., 2024), MARS with variance reduction (Yuan et al., 2024), and SOAP that apply AdamW updates to the Shampoo eigenbasis while amortizing the cost of eigendecomposition over multiple steps.

Recent benchmarking studies show that matrix-based optimizers with spectral preconditioning (e.g., Kron, Muon, Soap, etc.) generally outperform scalar-based ones (e.g., AdamW, Lion, Mars, etc.), though the optimal choice often depends on the specific scenario (Schlotthauer et al., 2025; Wen et al., 2025; Semenov et al., 2025). We are reminded that empirical studies cannot exhaustively cover all scenarios, even with optimal hyperparameters and experiment setup guidelines. Consequently, a promising research direction lies in characterizing the regimes where matrix-based spectral methods provide the most benefit, balancing their potential computational overhead against the sample efficiency and performance gains.

## 3 INSIGHTS ON SPECTRAL REGULARIZATION

In this section, we started by reviewing the spectral distributions in LLM pretraining and investigated the concept of principal regularization in mathematical function optimization.

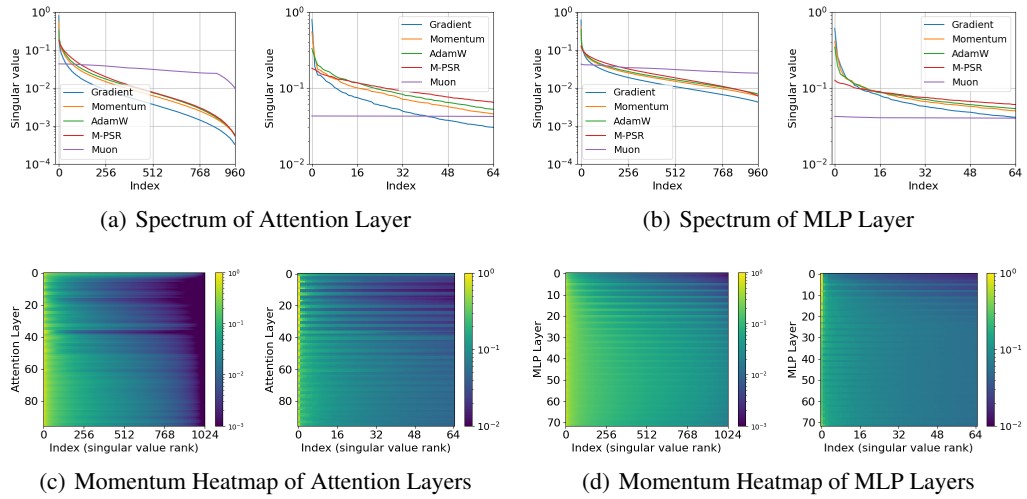

(a) Spectrum of Attention Layer      (b) Spectrum of MLP Layer

(c) Momentum Heatmap of Attention Layers    (d) Momentum Heatmap of MLP Layers

Figure 1: (Top) Spectral distributions of the final attention layer (a) and the final mlp layer (b). (Bottom) Heatmaps of spectral distributions across all attention (c) and mlp layers (d). Results are from an LLaMA-350M model at 1000 training steps on the C4/en dataset. All right-hand figures highlight the top 64 spectral values. Both gradient and momentum of all layers in the model exhibit a *spiked-head-heavy-tail* structure, where a few directions are dominant. Our proposed principal spectral regularization (PSR) method selectively orthogonalizes momentum along these dominant directions, balancing efficiency and computational cost.

### 3.1 A Spectral Perspective in LLM Pretraining

We begin by analyzing spectral distributions in pretraining LLMs. As illustrated by the spectral distributions of the final attention and MLP layer in Fig. 1, gradients are dominated by a few directions, followed by a heavy-tailed spectrum. We concluded our observation as a *spiked-head-heavy-tail* structure, where a few dominant directions (the *spiked-head*) capture most of the variance, while many others persist with smaller contributions. The momentum, as the running average of gradients, exhibits a similar structure but with fewer dominant directions, and *heavy-tail* lifted, allowing updates to focus more on important directions accordingly over training iterations. From the momentum spectral heatmaps of attention and mlp layers across a LLaMA-350M model (Fig. 1), a clear distinction arises between these attention layers (q, k, v, o) and MLP layers (gate, up, down): attention-layer momentum spectrum tend to decay more sharply, resembling a near $y = x$ slope in the tail and showing greater variability across layers, whereas MLP spectrum exhibit greater uniformity despite the same presence of dominant directions.

Adam produces updates that mirror the momentum spectrum but with attenuated spikes, using the second moment to adapt to local curvature. By plotting Gradient, Momentum, AdamW, and Muon sequentially, we observe that the latter methods increasingly diminish updates in the dominant directions while amplifying those in the tail. This trend continues until the update magnitudes become nearly uniform across spectral directions, with Muon achieving this by using Newton–Schulz iterations to approximate a semi-orthogonalization of the momentum. This is thought to promote exploration of those "rare but important directions" in the training process. Yet, such a uniform update may also amplify noisy directions and become unnecessary in such scenarios; in particular, the attention layers with momentum exhibited a rapidly decaying heavy-tail, as Adam predicted.

Viewing Adam's second moment and Muon's matrix orthogonalization as forms of momentum regularization, we hypothesize that full matrix regularization, considering the computational overhead, may not be necessary on all occasions. This paper investigates that possibility.

### 3.2 Principal Regularization in Mathematical Function Optimization

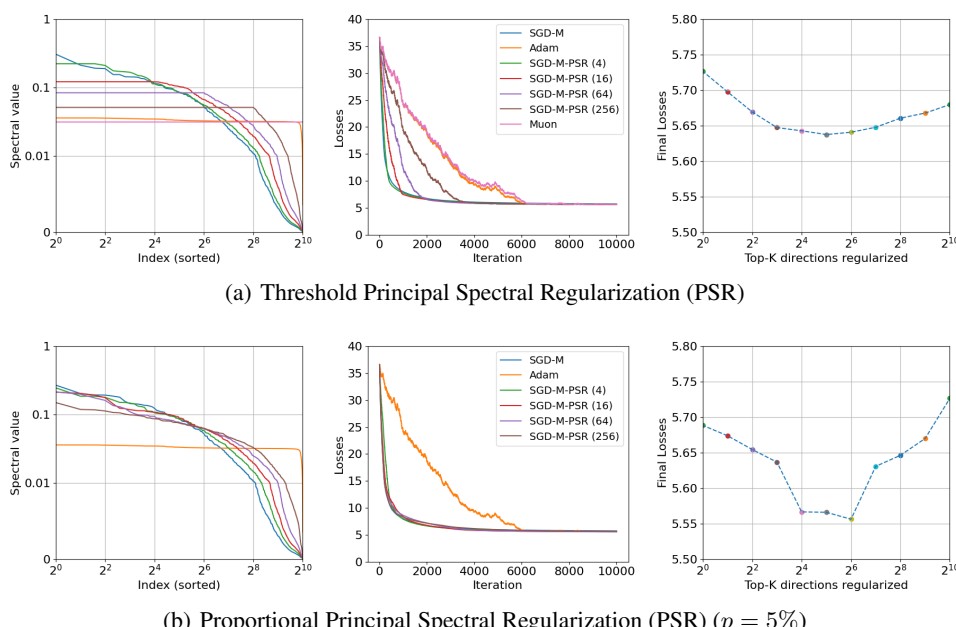

(a) Threshold Principal Spectral Regularization (PSR)

(b) Proportional Principal Spectral Regularization (PSR) ($p = 5\%$)

Figure 2: (Left) Update spectrum at step 1000, (Middle) Training dynamics, and (Right) Final loss for different numbers of regularized components of the Styblinski–Tang function ($n = 1024$) with weight attribution and Gaussian noise. We present two variants of the principal spectral regularization method: (a) shrinking the $K$ dominant updates to the top-$K$ threshold magnitude, and (b) scaling the $K$ dominant directions to a proportion of $p = 5\%$.

We start our investigation on a minimal benchmark of optimization, the Styblinski-Tang function defined in $n$-dimensional space, usually evaluated on the hypercube $x_i \in [-5, 5]$:

$$f(x) = \frac{1}{2} \sum_{i=1}^{n} \left( x_i^4 - 16x_i^2 + 5x_i \right) \tag{1}$$

Its global minimum $f(x^*) = -39.16599n$ can be found in $x^* = (-2.903534, \ldots, -2.903534)$. To better approximate realistic deep learning scenarios, we assign an additional weight to each dimension following a power-law distribution as $w_i = i^{-\alpha}$ to mimic the heavy-tailed spectrum commonly observed in neural network gradients or Hessians (Zhao et al., 2024a; Morwani et al., 2024; Tang et al.), and inject Gaussian noise with mean=0 and std=5e-3 at each step. We set the dimension of the function $n = 1024$ and use the baseline optimizer for SGD with momentum $m = 0.9$ and the learning rate $\eta = 0.01$, along with Adam with $(\beta_1, \beta_2) = (0.9, 0.95)$ for comparison.

In addition to SGD-momentum and Adam, we consider and evaluate two forms of partial spectral regularization approaches: (a) a threshold approach, which shrinks the $K$ dominant updates to the smallest magnitude among them, and (b) a proportional approach that scales the $K$ dominant updates by a fixed proportion $p$, as presented in Fig. 2. The threshold regularization approach with $K = n = 1024$, which enforces uniform updates in all directions, can be viewed as a replication of Muon. All update vectors are normalized to ensure a consistent step size across the different methods. We track loss over iterations and report the averaged final loss over multiple runs in Tab. 1.

| Optimization Method | $d$ | 4 | 16 | 64 | 256 | 1024 |
|---|---|---|---|---|---|---|
| SGD-M | 5.7264 | / | / | / | / | / |
| Adam | 5.6698 | / | / | / | / | / |
| SGD-M-PSR (Thr) | / | 5.6692 | 5.6426 | 5.6406 | 5.6605 | 5.6796 |
| SGD-M-PSR (20%) | / | 5.6629 | 5.5679 | 5.5775 | 5.6661 | / |
| SGD-M-PSR (10%) | / | 5.6561 | 5.5721 | 5.5610 | 5.6498 | / |
| SGD-M-PSR (5%) | / | 5.6543 | 5.5664 | **5.5561** | 5.6461 | / |
| SGD-M-PSR (1%) | / | 5.6618 | 5.6454 | 5.6576 | 5.7096 | / |

Table 1: Final loss on the Styblinski-Tang function ($n = 1024$) for Adam and SGD with momentum, with and without principal spectral regularization across different values of $d$.

The spectral distribution of Adam in Fig. 2 is nearly uniform, resembling the Muon spectrum in practice, with a sharp drop to zero in the final directions. Nevertheless, while both Adam and Muon converge to lower minima than SGD with momentum, their training trajectories are much slower and unstable. For principal spectral regularization, the fractional approach consistently outperforms the streamlined approach. In particular, the configuration with $p = 5\%$ and $d = 64$ achieves the lowest final loss among all tested optimizers, outperforming standard SGD-M, Adam, and Muon.

Moreover, the loss curves across different choices of $d$ reveal that there is an optimal trade-off between the number of directions being regularized and the strength of their penalization: too few directions lead to under-regularization, while too many may suppress useful update components. This finding suggests that uniform updates in parameters or spectral direction achieved by full matrix orthogonalization, adopted in Lion and Muon, may be unnecessary or even suboptimal in some scenarios. While these approaches explore "rare but important directions", they also risk amplifying noisy directions, which is especially relevant in the presence of noisy data or small-batch training. Additional results under more robust experimental settings are reported in Tab. 7, Appendix. 3.

## 4 PRINCIPAL SPECTRAL REGULARIZATION FOR LLM PRETRAINING

In this section, we propose a principal spectral regularization method for momentum-based optimizers tailored for high-dimensional matrices. According to our hypothesis that regularizing only a fraction of the dominant spectral components in the momentum can serve as an effective approximation to matrix orthogonalization, without explicitly altering the heavy tail, we approach this problem using matrix deflation along a set of identified spectral directions. While power iteration efficiently converges to the largest singular pair, applying it repeatedly to extract multiple adjoint singular directions through deflation becomes computationally prohibitive. To address this, we adopt

a simplified block Lanczos bidiagonalization procedure (or Golub-Kahan bidiagonalization), which combines power iteration with orthogonalization against both previous and current blocks. This allows us to identify a richer set of dominant directions in parallel. QR factorization, as the block orthogonalization technique, is employed at each step to maintain orthonormality and uniqueness of the approximate left and right singular vectors. By constructing a compact bidiagonal submatrix that captures the dominant spectral structure, we can deflate multiple principal directions in the momentum simultaneously, reducing computational overhead.

---

**Algorithm 1** Principal Spectral Regularization (PSR)

---

**Require:** Momentum $M \in \mathbb{R}^{(m \times n)}$, regularization factor $\eta$, Lanczos iteration $K$, rank $r$
1: $(U, B, V) \leftarrow \text{BIDIAGONAL}(M, K, r)$
2: $(U_b, *, V_b^\top) \leftarrow \text{SVD}(B)$
3: $u \leftarrow UU_b, \quad v \leftarrow V_b V^\top$
4: $M \leftarrow M - \eta\, u(u^\top M v^\top)v$
5: $M \leftarrow \text{NORMALIZE}(M)$
6: **return** $M$

---

1: **function** BIDIAGONAL($M, K, r$)
2:     $B \leftarrow 0 \in \mathbb{R}^{rm \times rm}$
3:     $u_0 \sim \mathcal{N}(0,1)^{N \times r}$
4:     $u_0, * \leftarrow \text{QR\_ORTHOGONAL}(u)$
5:     **for** $k = 0$ to $K - 1$ **do**
6:         $v_k \leftarrow M^\top u_k$
7:         $v_k, R_{\alpha,k} \leftarrow \text{QR\_ORTHOGONAL}(v_k, V\_blocks)$
8:         $B[rk : r(k+1), rk : r(k+1)] \leftarrow R_{\alpha,j}$
9:         **if** $k \geq K - 1$ **then**
10:           **break**
11:         **end if**
12:         $u_{k+1} \leftarrow Mv$
13:         $u_{k+1}, R_{\beta,k+1} \leftarrow \text{QR\_ORTHOGONAL}(u, U\_blocks)$
14:         $B[r(k+1) : r(k+2), rk : r(k+1)] \leftarrow R_{\beta,k+1}$
15:     **end for**
16:     $U \leftarrow [u_0, \ldots, u_{K-1}], \; V \leftarrow [v_0, \ldots, v_{K-1}]$
17:     **return** $U, B, V$
18: **end function**

---

1: **function** QR_ORTHOGONAL($Q$, prev_blocks)
2:     **for** each $Q_{\text{prev}} \in$ prev_blocks **do**
3:         $Q \leftarrow Q - Q_{\text{prev}}(Q_{\text{prev}}^\top Q)$
4:     **end for**
5:     $Q, R \leftarrow \text{QR}(Q)$
6:     **return** $Q, R$
7: **end function**

---

The proposed principal spectral regularization (PSR) method is presented in Alg. 1, in which the *bidiagonal* function can be viewed either as a block-wise adaptation of Power Iteration that produces orthonormal bases, or equivalently as an iterative randomized SVD method that incrementally captures dominant spectral components. By constructing two semi-orthonormal bases $U$ and $V$ and a reduced bidiagonal form $B$ of input momentum $M$ in the Krylov subspace, SVD is performed to extract singular vectors $(U_b, V_b)$ on $B$. The single vector groups are reconstructed as $u = UU_b$ and $v = V_b V^\top$, and matrix deflation is applied to $M$ with respect to $(u, v)$ using a regularization factor $\eta$. This procedure attenuates the dominant directions identified to $1 - \eta$ of their original magnitude. The deflated matrix is then normalized and rescaled to match the update scales of Muon and AdamW, as discussed below. By suppressing the dominant directions and normalization, the heavy-tailed spectrum is retained and relatively lifted, effectively amplifying their contributions.

| Method | A | PSR(A) | Newton-Schulz(A) | QR-Decomposition(A) |
|--------|---|--------|------------------|---------------------|
| $E_{\mathrm{ortho}}(B)$ | $8.0 \times 10^4$ | 32.0 | 9.9 | $2.4 \times 10^{-5}$ |
| $D_{\mathrm{sub}}(A, B)$ | / | $3.52 \times 10^{-5}$ | $3.51 \times 10^{-5}$ | $3.47 \times 10^{-5}$ |
| $D_{\mathrm{spec}}(A, B)$ | / | 0.082 | 0.209 | 0.367 |

Table 2: Matrix orthonormality $E_{\mathrm{ortho}}$, subspace distance $D_{\mathrm{sub}}$ and spectral fidelity $D_{\mathrm{spec}}$ scores between different matrix orthogonalization methods. For spectral fidelity evaluation, the input matrices are normalized to capture the shape difference only. PSR is less accurate than Newton-Schulz in matrix orthogonalization but achieves a comparable subspace distance and the lowest spectral fidelity, as it preserves the tail of the spectral distribution.

To verify our methodology, we evaluated and compared (1) orthonormality $E_{\mathrm{ortho}}(B) = \|B^\top B - I_r\|_F$, (2) subspace distance $D_{\mathrm{sub}}(A, B) = \|Q_B^\top Q_B - Q_A^\top Q_A\|_F$, and (3) spectral fidelity $D_{\mathrm{spec}}(A, B) = \frac{\|\sigma_A - \sigma_B\|_2}{\|\sigma_A\|_2}$ for singular values $\sigma$ and orthonormal basis $Q$, of our proposed regularization to Newton-Schulz and QR decomposition in random matrix $A \in \mathbb{R}^{(1024 \times 2048)}$. Tab. 2 demonstrated that PSR provides a less accurate orthogonalization of the input matrices compared to Newton–Schulz, yet achieves a subspace distance comparable to the other two methods. Its spectral fidelity is the lowest among the three methods, as PSR preserves the heavy-tail structure of the original spectrum mostly unchanged. The practical regularization effect is presented in **SGD-M-PSR** in Fig. 1, where the prominent *spiked-head* structure nearly disappears and the tail distribution is further elevated than with AdamW or plain momentum. In the following paragraph, we will discuss how PSR is integrated into SGD with momentum and its theoretical computational overhead.

**SGD Momentum with PSR:** We adopt a Nesterov-style momentum in SGD according to the empirical verifications of Muon and perform PSR on the lookahead gradient. The hyperparameters of PSR are configured according to the optimal setup in the Styblinski-tang function: optimal regularization factor $\eta = 0.95$, and Lanczos iteration $K = 2, r = \min(m, n)/32$ to regularize the top-1/16 spectral directions in momentum with adaptivity. Empirical results demonstrated that these constants provide an optimal balance between convergence performance and computational cost and behave consistently over different LLM scales and architectures.

**Update RMS Rescaling:** We introduce an extra scaling factor to align the update RMS with that of Muon and AdamW, thereby reducing the need for additional hyperparameter tuning. PSR normalizes the momentum by its $\ell_2$ norm at each step to an RMS of $1/\sqrt{mn}$. According to our empirical observations in Tab. 6 provided in Appendix. C, we multiply the momentum by $0.18\sqrt{mn}$ to align with update RMS with Muon and AdamW, followed by a parameter update with weight decay. The resulting SGD with PSR momentum is summarized in Alg. 2.

---

**Algorithm 2** SGD-Momentum with Principal Spectral Regularization (SGD-M-PSR)

---

**Require:** Weight $W_{t-1} \in \mathbb{R}^{m \times n}$ and Momentum $M_{t-1} \in \mathbb{R}^{m \times n}$ at step $t - 1, m \leq n$.
1: Initialize $M_t \leftarrow 0, t \leftarrow 0$.
2: **for** each step **do**
3:      $G_t \leftarrow \nabla \mathcal{L}_t(Wt - 1)$                                 ▷ Compute gradient
4:      $M_t \leftarrow \mu M_{t-1} + G_t$                            ▷ Accumulate momentum
5:      $\hat{G}_t \leftarrow \mu M_t + G_t$                       ▷ Nesterov lookahead gradient
6:      $\hat{O}_t \leftarrow \mathrm{PSR}(\hat{G}_t, K = 2, \eta = 0.5, r = \frac{m}{32})$      ▷ Principal Spectral Regularization
7:      $W_t \leftarrow W_{t-1} - \eta(0.18 \cdot \hat{O}_t \cdot \sqrt{mn} + \lambda W_{t-1})$     ▷ Update parameter with weight decay
8: **end for**

---

**Computational Complexity Analysis:** All operations in the PSR algorithm rely on low-rank matrix multiplications, effectively reducing the overall complexity to below cubic order w.r.t. momentum dimensions. The computational overhead of PSR is characterized by the following Theorem:

**Theorem 4.1** (Upper Bound of Computation Complexity of PSR). *For iteration number $K = 2$ and implicit rank $r = m/32$, the extra FLOPs required by PSR compared to SGD are at most $\mathcal{O}_{\mathrm{overhead}}(\mathrm{PSR}) < 1/2m^2n$ when the parameter dimension satisfies $16 \leq m \leq n$.*

The full proof and further discussion are provided in the Appendix. E. This overhead corresponds only to about $2\%$ of the extra FLOPs, the $30m^2n$ upper bound incurred by the 5-step Newton-Schulz in Muon (Jordan et al., 2024) for all LLMs and most deep learning models.

**Wall-clock time Comparison:** To fully exploit CUDA acceleration, we utilize the PyTorch API for QR factorization and SVD. As existing PyTorch linear algebra functions lack native half-precision arithmetic support with CUDA, explicit data-type conversions are required during mixed-precision training. Although theoretical analysis suggests that PSR introduces negligible computational overhead, in practice, it can be more time-consuming due to the sequential execution of iterative operations in small-scale LLMs, according to the runtime experiments in Fig. 3. Nevertheless, substantial gains in speed and memory efficiency are observed for large-scale models beyond 7B parameters and dimensions exceeding 4096, even under a naive PyTorch implementation. Future work will focus on kernelizing the Lanczos and deflation procedures to reduce the additional cost in training.

| Method | Params | Attention | | MLP | |
|---|---|---|---|---|---|
| | | Time | Memory | Time | Memory |
| **LLaMA-1.3B** | | (2048, 2048) | | (2048, 5461) | |
| NewtonSchulz | $T = 5$ | 2.01 (ms) | 64.1 (MB) | 4.68 (ms) | 120.0 (MB) |
| PSR (K=2) | $r = \frac{m}{32}$ | 4.85 (ms) | 18.5 (MB) | 4.44 (ms) | 48.2 (MB) |
| **LLaMA-3B** | | (2560, 2560) | | (2560, 6848) | |
| NewtonSchulz | $T = 5$ | 4.09 (ms) | 87.5 (MB) | 8.19 (ms) | 186.0 (MB) |
| PSR (K=2) | $r = \frac{m}{32}$ | 5.54 (ms) | 29.0 (MB) | 6.09 (ms) | 74.6 (MB) |
| **LLaMA-7B** | | (4096, 4096) | | (4096, 11008) | |
| NewtonSchulz | $T = 5$ | 14.83 (ms) | 224.0 (MB) | 30.22 (ms) | 472.0 (MB) |
| PSR (K=2) | $r = \frac{m}{32}$ | 9.63 (ms) | 74.1 (MB) | 11.77 (ms) | 189.6 (MB) |
| **LLaMA-70B** | | (8192, 8192) | | (8192, 16384) | |
| NewtonSchulz | $T = 5$ | 110.79 (ms) | 896.0 (MB) | 184.33 (ms) | 1536.0 (MB) |
| PSR (K=2) | $r = \frac{m}{32}$ | 29.93 (ms) | 296.5 (MB) | 35.85 (ms) | 568.5 (MB) |

Table 3: Comparison of orthogonalization methods on BFloat16 tensors. Each block presents Attention and MLP matrix shapes from a representative LLaMA model. Reported values denote the average over 1000 PyTorch runs, with peak GPU memory usage measured via TORCH.CUDA. Despite the overhead of sequential QR and SVD steps for low-dimensional cases, their reduced complexity yields significant runtime and memory gains on larger matrices in comparison to the Newton-Schulz iteration in a naive PyTorch setup.

## 5 EXPERIMENTS

In this section, we describe our experimental setups for LLM pretraining and present the results to analyze the impact of different spectral methods on training.

### 5.1 EXPERIMENT SETUP

In this paper, we follow the experimental setup described in Zhao et al. (2024a) and Raffel et al. (2020). We focus primarily on the LLaMA architecture with four varying sizes: $350M$, $1B$, $3B$, $7B$ parameters. The primary pre-training corpus is the C4/en dataset with a sequence length of $1024$. We consider a total batch size of $512$, and evaluate models at $10000$ training steps. Additionally, we trained a LLaMA-1.3B model until 36B tokens to assess the robustness of the optimizer in extended pretraining. The final model is evaluated on nine downstream benchmarks: ARC (Easy and Challenge), BoolQ, HellaSwag, OpenBookQA, PiQA, MMLU, WinoGrande, and SciQ. For comparison, we used standard SGD with Nesterov momentum (SGD-M) with Newton-Schulz iteration (Muon) or principal spectral regularization (SGD-M-PSR) alongside AdamW as the baseline. The hyperparameters are available in the Appendix. B.

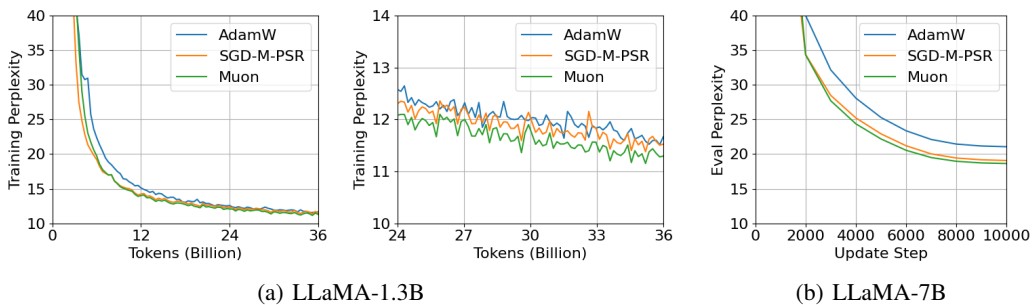

(a) LLaMA-1.3B                    (b) LLaMA-7B

Figure 4: (a) The LLaMA-1.3B model trained on 36B tokens, and (b) the LLaMA-7B model trained for $10,000$ steps on the `C4/en` corpus with a batch size of 1 per node. SGD-M-PSR converges the fastest during the warm-up phase, though the speed advantage diminishes in later stages of training.

## 5.2 EXPERIMENT RESULTS

Fig. 3 presents the training dynamics of three LLaMA architectures with four optimizers with different spectral regularization scales: SGD-M, Adam, SGD-M-PSR, and Muon. The experiment results indicate that while SGD with Nesterov-style momentum approaches AdamW on the LLaMA-350M model, it is unsuitable for training larger models due to its poor scalability. However, when PSR is applied, SGD-M consistently surpasses AdamW and even achieves a better validation perplexity than Muon on the LLaMA-1.3B model, with lower computational FLOPs overhead and reduced orthogonalization requirements. The results suggest that, under certain circumstances, regularizing only the leading spectral components in the momentum can achieve performance comparable to full-matrix orthogonalization, which updates all spectral directions with uniform magnitude.

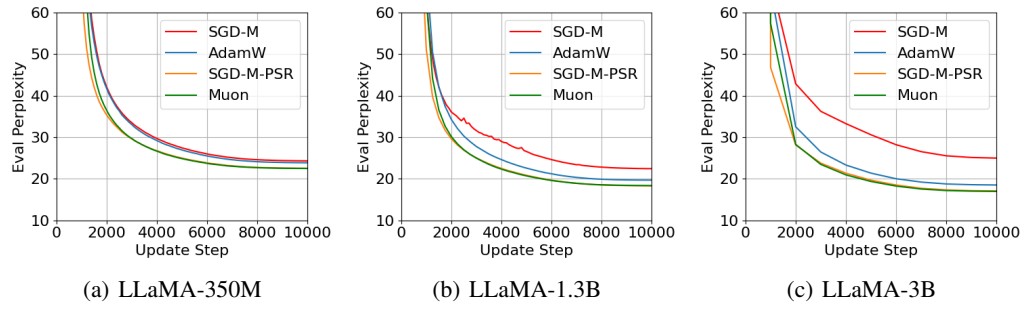

(a) LLaMA-350M                (b) LLaMA-1.3B                (c) LLaMA-3B

Figure 3: LLaMA-350M, 1.3B, 3B models trained $10000$ steps on the `C4/en` dataset with four different optimizers: SGD-M, SGD-M-PSR, AdamW, and Muon. SGD-M exhibits the most unstable pretraining dynamics among all optimizers, whereas SGD-M-PSR matches Muon in both sample efficiency and validation perplexity.

We further evaluate scaled training scenarios in Fig. 4: a LLaMA-1.3B model trained on 36B tokens with a total batch size of 2048, and a LLaMA-7B model with local batch size 1 and global batch size 512. For the LLaMA-1.3B model, SGD-M-PSR exhibits the most stable and rapid convergence during warm-up, but is eventually surpassed by Muon as training progresses. Nevertheless, its sample efficiency relative to AdamW remains consistently stronger in long-term training. The downstream evaluation in Tab. 4 also indicated that SGD-M-PSR outperforms AdamW on the commonly used language and knowledge benchmarks. On the other hand, for the LLaMA-7B model, SGD-M-PSR exhibits a small gap in comparison to Muon, yet still achieves a substantial speed-up over AdamW.

| Optimizer | ARC-e | ARC-c | BoolQ | HellaSwag | OBQA | PiQA | MMLU | WG | SciQ | Avg. |
|---|---|---|---|---|---|---|---|---|---|---|
| | LLaMA-1.3B [num_fewshot = 0] | | | | | | | | | |
| AdamW | 43.18 | 25.68 | 57.19 | 46.62 | 30.20 | 69.97 | 22.97 | **52.64** | 68.40 | 46.32 |
| SGD-M-PSR | 43.60 | **25.85** | **57.95** | 46.34 | 30.20 | **71.16** | 22.77 | **52.64** | 68.50 | 46.56 |
| Muon | **45.08** | 25.60 | 57.89 | **47.52** | **30.80** | 71.11 | **22.85** | 52.25 | **70.30** | **47.04** |
| | LLaMA-1.3B [num_fewshot = 2] | | | | | | | | | |
| AdamW | 47.26 | 25.60 | 50.31 | 46.43 | **32.40** | 69.70 | **25.05** | 52.41 | 77.80 | 47.44 |
| SGD-M-PSR | 48.11 | 26.54 | 53.09 | 46.48 | 30.60 | 70.46 | 24.32 | **53.28** | 77.80 | 47.86 |
| Muon | **49.24** | **27.05** | **56.88** | **47.34** | 31.60 | **71.38** | 24.39 | 51.62 | **78.10** | **48.62** |

Table 4: A comparison of average downstream task performance in 0-shot and 2-shot settings of different optimizers on a LLaMA-1.3B model trained with 36B tokens on the `C4/en` corpus. WG denotes WinoGrande. SGD-M outperforms AdamW with PSR across most benchmarks while falling short in comparison to Muon.

From the over-trained experiment on LLaMA-1.3B, we observe and report that SGD-M with PSR is still stricly worse than running Muon, representing a poorer converging ability in the loss-steady training stage. We hypothesize that this is due to Muon's full matrix orthogonalization, which further amplifies and stabilizes the low-magnitude update directions, in the heavy-tail distribution, that PSR strengthened relatively less. By preserving updates in these weak directions, Muon appears better equipped to navigate and settle into lower-loss regions of the landscape. Despite this, the proposed PSR approach demonstrates the optimization trade-offs behind head-only shrinkage and complete normalization, revealing the different roles played by the head and tail components in the momentum spectra. We report further ablation studies to analyze the scalability of partial orthogonalization on optimization performance in the Tab. 9, Appendix. D.

## 6 CONCLUSION

In this paper, we analyze the preconditioning by Momentum, Adam, and Muon through a universal spectral regularization perspective. While prior regularization and optimization methods primarily focus on low-rank projection-based methods to approximate the ideal preconditioner, we propose a marginal approach that targets a small subset of spectral directions in the original parameter space. By selectively penalizing the dominant directions in the momentum, our proposed PSR method enables SGD with momentum to surpass Adam in large-scale LLM pretraining. Although our PSR method does not match Muon in downstream performance or scaled-up training, SGD-M-PSR consumes only 2% of the extra FLOPs required by the Newton-Schulz iteration in theoretical complexity compared to standard SGDs, highlighting a promising direction for next-generation spectrum-aware optimizer design. On the other hand, the geometric property of PSR also benefits future empirical analysis in understanding the roles of different spectral components, enabling an in-depth inspection of the critical design choices behind AdamW, Muon, and other optimizers. Future work can expand this line of inquiry by developing more principled mathematical frameworks and more efficient partial orthogonalization strategies.

## ETHICS STATEMENT

This paper aims to understand the foundation of deep learning optimization. While it may have many potential societal consequences, we think none of them must be specifically discussed here.

## REPRODUCIBILITY STATEMENT

All experiments in this paper were conducted using publicly available datasets and models. We provide the detailed training configurations in the Section. 5.1 and Appendix. B, including model architecture, optimization parameters, and learning rate schedules. The optimizer implementation, along with scripts for data preprocessing, training, and evaluation, will be released upon publication.

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

## A   STATEMENT ON THE USE OF LLMs

In preparing this manuscript, LLMs (mostly GPT-4/5) are utilized for linguistic refinement, including the detection and correction of grammar errors or spelling mistakes, and sentence rephrasing to improve clarity, coherence, and readability. LLMs were also referenced when structuring the paper contents and reviewing missing details, but were not involved in the formulation of ideas, the execution of experiments, or the generation of experimental results in this article.

## B   HYPERPARAMETERS

In this paper, we follow the experimental setup described in Zhao et al. (2024a) and Raffel et al. (2020). The model architecture and respective hyperparameters are presented in Tab. 5. Besides the learning rate and batch size settings, we use a cosine decay learning rate scheduler with a minimum learning rate ratio of 0.1 for all experiments. Weight decay is set to 0.1, and gradients are clipped at 1.0. The LLaMA models are tokenized using the T5 tokenizer, and all training is performed in *BFloat16* mixed precision.

| Model | LLaMA-350M | LLaMA-1.3B | LLaMA-3B | LLaMA-7B | LLaMA-1.3B |
|---|---|---|---|---|---|
| Layer num | 24 | 24 | 32 | 32 | 24 |
| Hidden dim size | 1024 | 2048 | 2560 | 4096 | 2048 |
| FFN dim size | 2736 | 5461 | 6848 | 11008 | 5461 |
| Attention heads | 16 | 32 | 32 | 32 | 32 |
| Seq-len | 1024 | | | | 2048 |
| LR | $3.0 \times 10^{-4}$ | $3.0 \times 10^{-4}$ | $3.0 \times 10^{-4}$ | $3.0 \times 10^{-4}$ | $3.0 \times 10^{-4}$ |
| Batch Size | 8 | | | 1 | 8 |
| GradAcc | 8 | | | 64 | 1 |
| Total Batch Size | 512 | | | | 2048 |
| Iterations | 10000 | | | | 9000 |
| Warmup iterations | 1000 | | | | 2000 |

Table 5: Training configurations for different LLaMA model scales, including architecture details, sequence length, learning rate, batch size and gradient accumulation steps, and training schedule.

## C   UPDATE RESCALING

Our empirical observation in Tab.6 reported that AdamW and Muon's update RMS falls in the range of [0.15, 0.2]. AdamW shows lower RMS during the warm-up stage but higher at later stages, while Muon's update RMS is lower than 0.2 on average. According to these insights, we opt to set the rescaling factor to 0.18 for aligning update RMS with AdamW and Muon in our experiment setups.

| Update Step | Update RMS | Adam | Muon | SGD-M-PSR |
|---|---|---|---|---|
| 1000 | Attention Avg. | $1.26 \times 10^{-1}$ | $1.53 \times 10^{-1}$ | |
| | MLP Avg. | $1.58 \times 10^{-1}$ | $1.77 \times 10^{-1}$ | $1.8 \times 10^{-1}$ |
| 2000 | Attention Avg. | $1.54 \times 10^{-1}$ | $1.63 \times 10^{-1}$ | |
| | MLP Avg. | $1.88 \times 10^{-1}$ | $1.79 \times 10^{-1}$ | |

Table 6: The update RMS of the three different optimizers at 1000 and 2000 training steps of a LLaMA-350M model on the `C4/en` dataset. The update RMS is averaged respectively from all Attention layers and all MLP layers. As we observe AdamW and Muon update RMS falls below 0.2, we adopt a rescaling factor of 0.18 in SGD-M-PSR for alignment.

# D  ABLATION STUDIES AND ADDITIONAL RESULTS

In this section, we present further ablation studies and additional results as an extension to the main paper, further evaluating and comparing different optimizer settings. The results for mathematical function optimization and LLM pretraining are presented in the following subsections, respectively.

## D.1  ADDITIONAL RESULTS IN MATHEMATICAL FUNCTION OPTIMIZATION

In the main paper, we only considered a single case of the Styblinski-Tang function optimization problem. Tab. 7 presents the final loss and standard deviation results from repeated experiments through different random initialization, under different weight distribution and noise settings. We have reduced the function dimension and training iterations to 2000 to reduce the time for repeated experiments. We compare Adam to the proportional PSR approach, with parameters top-$K = n/16 = 16$ and proportion $p = 5\%$, which are optimal parameters in both Styblinski-Tang function optimization and pretraining LLMs. While the connection between mathematical function optimization and LLM pretraining is relatively vague, this toy problem served the purpose of demonstrating the motivation of PSR and the hypothesis that full orthogonalization can be sub-optimal.

| Setup | | Power-law Weight $w_i = i^{-\alpha}$, Noise $\epsilon \sim \mathcal{N}(\mathbf{0}, \sigma^2)$ | | |
|---|---|---|---|---|
| **Method** | **Final Loss** | $\alpha = 0.8$ | $\alpha = 1.5$ | $\alpha = 2$ |
| Adam | $\sigma = 0.001$ | $6.849 \pm 3.160$ | $6.875 \pm 3.174$ | $6.850 \pm 3.162$ |
| SGD-M-PSR | | $6.728 \pm 3.056$ | $6.703 \pm 3.147$ | $6.739 \pm 3.069$ |
| Adam | $\sigma = 0.01$ | $6.742 \pm 3.135$ | $6.772 \pm 3.109$ | $6.839 \pm 3.091$ |
| SGD-M-PSR | | $6.638 \pm 3.101$ | $6.665 \pm 3.086$ | $6.694 \pm 3.114$ |

Table 7: Final loss and standard deviation obtained when optimizing the Styblinski-Tang function ($n = 256$) with Adam and SGD-M-PSR, averaged across 1000 random initializations in the range $[-5, 5]$, and seed in the range $[1, 1000]$. The training iterations are set to 2000 and a learning rate of 0.05. For PSR, principal spectral regularization, we choose the proportional approach with top-$K = n/16 = 16$ and $p = 5\%$, which shrinks the top-16 update directions to 5%, aligning with the parameters used in LLM pretraining experiments.

## D.2  ABLATION STUDIES IN LLM PRETRAINING

Tab. 8 presents the downstream task evaluation on the LLaMA-3B and LLaMA-7B model (reported in Fig. 3), under a 0-shot setting. All models are trained with 2B tokens on the `C4/en` corpus. We observe that in the early stage of LLM pretraining (2B tokens), larger models (7B) underperform smaller models (3B), suggesting slower convergence. The relative ranking of optimizers remains consistent with the results on LLaMA-1.3B (Tab. 4): Muon performs best, followed by SGD-M-PSR, and then AdamW in compressive evaluation. We hope future work will extend these experiments to later training stages.

| Optimizer | ARC-e | ARC-c | BoolQ | HellaSwag | OBQA | PiQA | MMLU | WG | SciQ | Avg. |
|---|---|---|---|---|---|---|---|---|---|---|
| | LLaMA-3B [num_fewshot = 0] | | | | | | | | | |
| AdamW | 17.75 | 42.30 | 57.00 | 29.76 | **23.02** | **16.40** | 64.80 | 71.70 | 50.36 | 41.45 |
| SGD-M-PSR | **19.45** | **46.17** | 58.26 | **30.79** | 22.92 | 15.00 | 64.74 | **74.90** | 51.07 | 42.59 |
| Muon | 19.11 | 45.29 | **61.90** | 30.64 | 22.98 | **16.40** | 65.67 | 73.10 | **51.62** | **42.97** |
| | LLaMA-7B [num_fewshot = 0] | | | | | | | | | |
| AdamW | 17.92 | 42.34 | 61.47 | 29.14 | **23.13** | 15.00 | 62.95 | 68.10 | 51.07 | 41.23 |
| SGD-M-PSR | **18.69** | 43.06 | **62.26** | 29.80 | 22.90 | 16.60 | 64.53 | 71.30 | 50.83 | 42.22 |
| Muon | 18.26 | **43.90** | 59.54 | **29.95** | 23.03 | **17.60** | **64.69** | **74.90** | **51.85** | 42.64 |

Table 8: A comparison of average downstream task performance in 0-shot settings of different optimizers on a LLaMA-3B and a LLaMA-7B model trained with 2B tokens on the C4/en corpus. WG denotes WinoGrande. SGD-M outperforms AdamW with PSR across most benchmarks while falling short in comparison to Muon.

Table.9 presents the ablation study of SGD-M-PSR under different hyperparameter settings alongside other optimizers. We reported that in terms of final test perplexity, an optimal regularization factor of $\eta = 0.95$ improves performance consistently across all models as the selected principal components are punished to 5%.A larger regularization factor can introduce instability in the optimization process, as excessive head-only shrinkage may distorts the overall update directions. On the other hand, the choice of the rank-proportion coefficient $m/r$ reveals a computational trade-off between cost and efficiency: penalizing a larger fraction of spectral components improves sample efficiency, but at the expense of increased computational and memory cost per step. This observation highlights a scalable perspective in spectral preconditioning and points towards the efficiency of geometric-adaptive methodologies.

| Model | | | LLaMA-350M | LLaMA-1.3B |
|---|---|---|---|---|
| Optimizer | $1 - \eta$ | $m/r$ | Final Test Perplexity | |
| SGD-M | | | 24.31 | 22.44 |
| Adam | | | 23.85 | 19.69 |
| SGD-M-PSR | 5% | 128 | 23.72 | 18.88 |
| SGD-M-PSR | 10% | 32 | 23.49 | 18.52 |
| SGD-M-PSR | 5% | 64 | 23.54 | 18.50 |
| SGD-M-PSR | 7.5% | 32 | 23.36 | 18.42 |
| SGD-M-PSR | 5% | 32 | 22.54 | **18.30** |
| Muon | | | **22.49** | 18.36 |

Table 9: A comparison of final test perplexity of SGD-M-PSR with different hyperparameter settings on a LLaMA-350M and a LLaMA-1.3B model trained 2B tokens on the C4/en corpus. The regularization factor $\eta$ is chosen from $\{0.9, 0.925, 0.95\}$ with the rank proportion coefficient $m/r$ chosen from $\{32, 64, 128\}$. We observe that an appropriate regularization factor can slightly improve training performance, whereas increasing the proportion of principal components trades higher per-step computational cost for improved sample efficiency.

## D.3 COMPARISON WITH SOAP

Low-rank projection methods are widely used for approximating high-dimensional gradients, momentum, or second-order preconditioners in the field of optimization, including GaLore (Zhao et al., 2024a), Shamoo (Gupta et al., 2018), SOAP (Vyas et al., 2024), etc. These methods often compute and maintain a low-rank subspace and project future computations into that subspace, reducing the computational cost. Therefore, the optimization dynamics often happen in an approximated geometry of the full parameter space. In comparison, PSR identifies the principal spectral components in the original geometric parameter space, without maintaining any low-rank projectors or preconditioners across update steps/iterations.

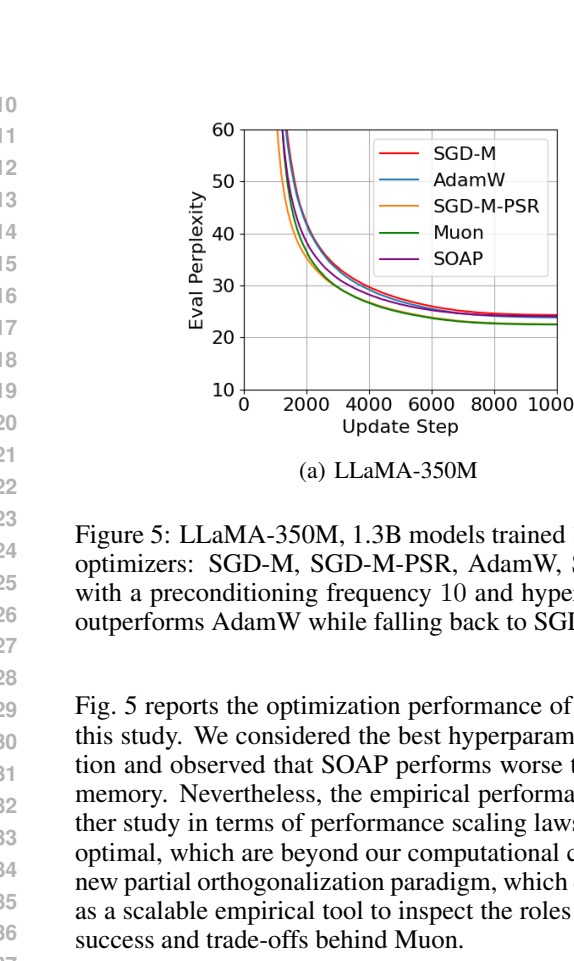

(a) LLaMA-350M             (b) LLaMA-1.3B

Figure 5: LLaMA-350M, 1.3B models trained 10000 steps on the `C4/en` dataset with five different optimizers: SGD-M, SGD-M-PSR, AdamW, SOAP, and Muon. In our experiment setup, SOAP with a preconditioning frequency 10 and hyperparameters $\beta_1 = 0.95, \beta_2 = 0.99, \beta_{\text{shampoo}} = 0.99$ outperforms AdamW while falling back to SGD-M-PSR and Muon.

Fig. 5 reports the optimization performance of SOAP in comparison to the optimizers discussed in this study. We considered the best hyperparameters reported in Vyas et al. (2024) in our reproduction and observed that SOAP performs worse than SGD-M-PSR while also consuming more GPU memory. Nevertheless, the empirical performance of the so-far mentioned optimizers requires further study in terms of performance scaling laws across different architectures trained to Chinchilla-optimal, which are beyond our computational capabilities. The purpose of our study focuses on the new partial orthogonalization paradigm, which differs from SOAP principally, and the power of PSR as a scalable empirical tool to inspect the roles of different spectral components and understand the success and trade-offs behind Muon.

# E    COMPUTATIONAL COMPLEXITY ANALYSIS

In this section, we present our proof of Theorem. 4.1 and discuss the overhead with Muon.

**Proof of Theorem. 4.1**

*Proof.* We first analyze the computational cost of PSR by accounting for the FLOPs of each component in the PSR regularization method:

- **QR-Orthogonal:**

    - **Orthonormal Projection:** For the previous block input of length $k$, the iterative projection cost is $k(4mr^2 + mr)$ or $k(4nr^2 + nr)$.
    - **QR Decomposition:** The classical (Householder) QR decomposition has complexity in $2AB^2 - 2B^3/3$ FLOPs for matrix of shape $[A, B]$ and $A > B$ (Golub & Van Loan (2013), §5.2.9). Here in the QR-Orthogonal function with input $Q$ of shape $[m, r]$ or $[n, r]$, it cost $2mr^2 - 2r^3/3$ or $2nr^2 - 2r^3/3$, upper bounded by $2mr^2$ or $2nr^2$.

- **Bi-Diagonal:** The Bi-Diagonal function procedure executed two operations: the QR-Orthogonalization, and power iteration. The double-sided matrix-vector multiplications contribute $(2K - 1)mnr$ FLOPs in $K$ iterations. The accumulative FLOPs of QR-Orthogonalization are characterized by $2K(m + n)r^2 + \frac{K(K-1)}{2}(4mr^2 + mr) + \frac{K(K-1)}{2}(4nr^2 + nr) = 2K^2(m + n)r^2 + \frac{K(K-1)}{2}(m + n)r$.

- **Principal Spectral Regularization:**

    - **Bidiagonalization:** The complete FLOPs for $K$ steps bidiagonalization is $2K^2(m + n)r^2 + \frac{K(K-1)}{2}(m + n)r + (2K - 1)mnr$

- **SVD:** The classical LAPACK SVD has complexity in $4AB^2 + 8AB^2 + 9B^3$ FLOPs for matrix of shape $[A, B]$ and $A > B$ if both singular vectors $U$ and $V$ are required (Golub & Van Loan (2013), §8.6.3). For our bidiagonal matrix $B$ with shape $[Kr, Kr]$ as input, computing its SVD requires $21(Kr)^3$ FLOPs.
- **Matrix Deflation:** The deflation step requires computing four matrix-vector products and two matrix-wise scalar operations in $4mnr + 2mn$ FLOPs.
- **Normalization:** The normalization contribute $2mn$ FLOPs for matrix in $[m, n]$.

Summing the above contributions yields the total overhead $\mathcal{O}_{\text{overhead}}(\text{PSR}) = 21(Kr)^3 + 2K^2(m+n)r^2 + \frac{K(K-1)}{2}(m+n)r + (2K+3)mnr + 4mn$.

For $K = 2$ and $r = m/32$, we have

$$\mathcal{O}_{\text{overhead}}(\text{PSR}) = 168r^3 + 8(m+n)r^2 + (m+n)r + 7mnr + 4mn$$

$$= \frac{168}{2^{15}}m^3 + \frac{1}{2^7}(m^3 + m^2n) + \frac{7}{2^5}m^2n + \frac{1}{2^5}(m^2 + mn) + 4mn$$

$$= \frac{53}{4096}m^3 + \frac{29}{128}m^2n + \frac{129}{32}mn + \frac{1}{32}m^2$$

We then derive the condition for the target inequality to hold:

$$\frac{53}{4096}m^3 + \frac{29}{128}m^2n + \frac{129}{32}mn + \frac{1}{32}m^2 \leq \frac{1}{2}m^2n$$

$$\frac{53}{8}m^3 + 2064mn + 16m^2 \leq 140m^2n$$

$$\frac{53}{8}\frac{m}{n} + 2064\frac{1}{m} + 16\frac{1}{n} \leq 140$$

According to our matrix dimension assumption that $m \leq n$, we may derive:

$$\frac{53}{8}\frac{m}{n} + 2064\frac{1}{m} + 16\frac{1}{n} \leq \frac{53}{8}\frac{m}{m} + 2064\frac{1}{m} + 16\frac{1}{m} \leq 140$$

$$\frac{53}{8} + 2080\frac{1}{m} \leq 140$$

$$2080\frac{1}{m} \leq \frac{1067}{8} \Rightarrow m \geq 15.6$$

In summary, under the condition $n \geq m \geq 16 > 15.6$, we have $\mathcal{O}_{\text{overhead}}(\text{PSR}) \leq \frac{1}{2}m^2n$. □

**Discussion:** We notice that condition $160 \leq m \leq n$ holds for all LLMs, including LLaMA-20M ($m = 256$) and GPT2-small ($m = 768$), as well as most deep learning architectures. Compared with the additional FLOPs required by Muon, bounded by $30m^2n$ for 5 Newton-Schulz iterations, PSR reduces the computational overhead to approximately 2%. Considering the baseline FLOPs to perform a single forward-backward step of training on a linear layer is $6mnB$, where B is the number of inputs, which is the batch size in tokens for LLMs; The FLOP overhead of PSR is $\frac{m}{12B}$, for parameter dimension $m$. We now calculate the overhead for two concrete training scenarios as follows Jordan et al. (2024):

- For GPT2-small with model dimension $m = 768$ and the example number of tokens per batch $B = 524288$, the overhead of PSR is $1.2 \times 10^{-4}$.
- For LLaMA-405B training with $m = 16384$ and tokens per batch $B = 1.6 \times 10^7$, the overhead of PSR is $8.4 \times 10^{-5}$.

Although the theoretical analysis suggests that PSR introduces negligible computational overhead, in practice, it can be more time-consuming due to the sequential execution of iterative operations. Future work will focus on kernelizing the Lanczos and deflation procedures to further reduce this additional cost during training.

