# OpenReview forum: "Principal Spectral Regularization Makes Momentum Surpass Adam for LLM Training"
_ICLR.cc/2026/Conference — Submitted to ICLR 2026_

### Official Review · Reviewer_yyW5 · 2025-10-27

**Soundness:** 2
**Presentation:** 2
**Contribution:** 2
**Rating:** 4
**Confidence:** 4

**Summary:**

This paper proposes a lightweight spectral method that allows standard momentum (SGD-M) to outperform Adam in large language model pretraining. The authors introduce Principal Spectral Regularization (PSR), which selectively penalizes only a few dominant spectral directions in the momentum, avoiding the heavy computation of full orthogonalization used by Muon. Experiments on LLaMA models show that PSR achieves similar or better performance than Adam and even approaches Muon’s efficiency, while using only about 2% of its additional FLOPs

**Strengths:**

The idea is well motivated and solution is clear and straightforward

**Weaknesses:**

1. Figure 3 the experiments are not trained to Chinchilla optimal, only 10K steps where perplexity is relatively high. Recent paper [1] has shown a lot of the optimizer papers' claim or not necessarily solid under closer inspection.
2. Figure 4 shows on 1.3B, PSR is strictly worse than running Muon.
3. No wall-clock improvement shown in real training runs. The series of operations proposed is not necessarily faster than matrix multiplication on GPU.
[1] Wen, Kaiyue, et al. "Fantastic pretraining optimizers and where to find them." arXiv preprint arXiv:2509.02046 (2025).

**Questions:**

1. Has learning rate and other hyperparameters been swept? If it has, can you specify the methodology you use to decide them (3.0 × 10−4, and 0.1)?

2. Setting the rescaling factor to 0.18 seems odd, since only the top directions are normalized, which means the overall update is in general smaller than Muon, where Muon has all eigenvalues set to 1? Shouldn't it be a larger scaling factor than 0.2 instead?

---

> ### Author Response · Authors · 2025-11-21
>
> We thank the reviewer for the thoughtful and constructive feedback. We are grateful for this opportunity, which enables us to clarify and discuss the critical questions pointed out by the reviewers, improve our paper quality. We have updated our manuscript with improved writing details and additional results, continuing to supply during the rebuttal period (due to limited computational resources and the length of pretraining experiments). Response to each weakness and question is listed below.
>
> **W1. Figure 3 experiments are not trained to Chinchilla optimal**: As discussed in Appendix.B, the experimental setup (and hyperparameters) is followed from [1] and [2], which are standard training schemes for comparing optimizers. The Chinchilla-optimal experiment is shown in Fig.4, where LLaMA-1.3B is trained on 36B tokens, 1.4x compute optimal.
>
> **W2. Figure 4 shows that on 1.3B, PSR is strictly worse than running Muon**: In an over-trained setting, SGD-M with PSR is strictly worse than running Muon, representing a poorer converging ability in the final/loss-steady training stage. We hypothesize that this is due to Muon's full matrix orthogonalization, which further amplifies and stabilizes the low-magnitude update directions (in the heavy-tail distribution) that PSR strengthened relatively less. By preserving updates in these weak directions, Muon appears better equipped to navigate and settle into lower-loss regions of the landscape. Despite this, the FLOPs overhead by PSR are far less than the Newton-Schula iteration utilized in Muon, where the real-world computational costs are also relatively low for large LLaMA models, as demonstrated by the wall-clock and memory comparison in Tab.3 we added in the updated manuscript. We believe our study exhibits a demonstration of how less computation in optimizer preconditioning may match advanced optimizers like Muon.
>
> **W3. The series of operations proposed is not necessarily faster than matrix multiplication on GPU**: We have updated the manuscript with Tab.3 on page 8, demonstrating that even under a naive PyTorch implementation, the gains in runtime and memory on large-scale LLaMA layers are significant, representing a promising direction for designing new optimizers. However, the runtime on smaller models is indeed more time-consuming due to the optimization benefits of matrix multiplication. We are looking forward to future refinements, including kernel attempts and community suggestions to improve the algorithm.
>
> **Q1. Have learning rate and other hyperparameters been swept?**: The experiment hyperparameters, including the learning rate, follow the setup in [2]. Computational resource limitations prevent us from conducting further or more thorough overtrained experiments. We are reminded by Wen, Kaiyue, et al that empirical studies cannot exhaustively cover all scenarios, even with optimal hyperparameters and experiment setup guidelines. Despite this, we believe that the idea of penalizing only a proportion of spectral update directions is worthwhile to explore for the machine learning community, including the development of more principled mathematical frameworks and code implementations. The contribution of our study is to re-examine the basic concepts and highlight a potential direction for future studies, and to strike a balance between computational overhead against sample efficiency and performance gains.
>
> **Q2. Setting the rescaling factor to 0.18 seems odd**: The update rescaling is discussed in Appendix.C, where we reported that the update RMS of AdamW and Muons falls in the range of $[0.15, 0.2]$, as shown by Tab.6. According to these insights, we opt to set the rescaling factor to 0.18 for aligning update RMS with AdamW and Muon. From a theoretical perspective, Kimi-AI proposed that effective rank usually equals $r=\min(n,m)$ for $RMS(O_t)=\sqrt{r/mn}$, thus setting the rescaling factor to $0.2\sqrt{\max (n, m)}$ in their Muon. Despite this, the effective rank of model layer matrices could fall short of that assumption, resulting in a slightly lower norm in practice (<1), which is why we set the rescaling factor to 0.18 after normalizing the update matrix in PSR (norm strictly equals 1 after normalization).
>
> We would like to sincerely thank the reviewer for their patience and professionalism demonstrated in the review process. We are willing to clarify the details that remain unclear in our study, and we are looking forward to future constructive discussions with the reviewer.
>
> [1]. Colin Raffel, Noam Shazeer, Adam Roberts, Katherine Lee, Sharan Narang, Michael Matena, Yanqi Zhou, Wei Li, and Peter J Liu. Exploring the limits of transfer learning with a unified text-to-text transformer. Journal of machine learning research, 21(140):1–67, 2020.
>
> [2]. Jiawei Zhao, Zhenyu Zhang, Beidi Chen, Zhangyang Wang, Anima Anandkumar, and Yuandong Tian. Galore: Memory-efficient llm training by gradient low-rank projection. arXiv preprint arXiv:2403.03507, 2024a.

---

### Official Review · Reviewer_we5J · 2025-10-28

**Soundness:** 2
**Presentation:** 2
**Contribution:** 1
**Rating:** 2
**Confidence:** 3

**Summary:**

In this paper, the authors introduced Principal Spectral Regularization (PSR). It changes the standard momentum updates to focus only on a few dominant spectral directions. The authors have shown that using PSR and SGD with momentum can have a better performance than Adam in LLM pretraining.

**Strengths:**

The strengths of this paper are summarized as follows:

1. The authors have given strong empirical evidence, showing a consistent improvement over AdamW. The authors have tested the LLaMA architecture with multiple sizes, 350M, 1B, 3B, and 7B parameters. It covers both small and large models.

2. This paper is easy to follow and easy to understand. LLM pretraining is an important area of research, and I believe that this paper will be interesting to the machine learning community.

**Weaknesses:**

The weaknesses of this paper are summarized as follows:

1. This paper lacks a theoretical analysis. The paper does not include a formal and rigorous convergence proof. The analysis is mainly empirical. It might not be so convincing whether or not the empirical performance can be generalized to other models.

2. For the experimental results, this paper lacks evaluation on a more diverse architecture. This paper only focuses on the LLaMA model and the C4 dataset. Also, it would be better to include an ablation study on the number of spectral components per layer.

3. Another minor comment is that the authors wrote “Fig. ??”. I think this should be a typo, and the author wants to refer to Figure 1?

**Questions:**

Please see the weaknesses.

---

> ### Author Response · Authors · 2025-11-21
>
> We thank the reviewer for the thoughtful and constructive feedback. We are grateful for this opportunity, which enables us to clarify and discuss the critical questions pointed out by the reviewers, improve our paper quality, and potentially contribute to the field. We have updated our manuscript with improved writing details and additional results, continuing to supply during the rebuttal period (due to limited computational resources and the length of pretraining experiments). Response to each weakness (question) is listed below.
>
> **W1. This paper lacks a theoretical analysis**: The reviewer's comments are sincerely appreciated. However, conducting theoretical studies in the optimization mechanisms of cutting-edge large-scale neural architectures remains difficult and is beyond our existing capabilities. We are open to any suggestions on conducting theoretical studies on both toy problems and LLM optimization to further strengthen our study. Apart from the weak theoretical perspective, we wish to clarify that the contribution of this study lies in the demonstration and comparison of gradient, momentum, and update matrices by AdamW and Muon from a continuous geometric perspective, and the identification of the empty gap in partial momentum preconditioning, which remains unexplored in the field. By implementation and validation of the PSR, we successfully bridged the gap by presenting the methodology and conclusions, which may inspire next-generation optimizers to achieve more optimal computational trade-offs than AdamW and Muon. We believe the concept of penalizing only a proportion of spectral update directions and partial matrix orthogonalization is worthwhile to explore, and kindly ask the reviewer to consider the potential insights this work may offer to the deep learning community.
>
> **W2. This paper lacks evaluation on a more diverse architecture**: We are managing to run experiments on the NanoGPT and OpenWebText benchmark and are expecting to supply the additional results and further ablation studies during this rebuttal period for continued discussion. Despite this, we believe the value and contribution of our study lies in re-examining optimizer preconditioners from a uniform geometric regularization perspective, and emphasize the potential of partial orthogonalization in the computation efficiency of large-scale models.
>
> **W3. The authors wrote ``Fig.??''**: The LaTeX error is now fixed in the updated manuscript. The reviewer's kind suggestions are greatly appreciated.
>
> We would like to sincerely thank the reviewer for their patience and professionalism demonstrated in the review process. We are looking forward to constructive discussions with the reviewer, and to when additional experiment results can be appended to this study.

---

> > ### Comment · Reviewer_we5J · 2025-11-26
> > **Thank you for your response!**
> >
> > I acknowledge that I have read the responses from the authors. My concerns are partially addressed, but the concern about the lack of a theoretical contribution remains. I choose to maintain my original rating.

---

### Official Review · Reviewer_5xga · 2025-10-30

**Soundness:** 2
**Presentation:** 3
**Contribution:** 2
**Rating:** 4
**Confidence:** 3

**Summary:**

The paper proposes principal spectral regularization for optimizer updates, arguing that attenuating only the few strongest momentum directions can suffice for effective LLM training. It claims that full momentum orthogonalization, as in Muon, is often unnecessary and may be suboptimal for LLM pretraining given the computational cost of Newton–Schulz and potential amplification of tail noise. The evidence combines spectral analyses, a controlled function-optimization study, and LLM pretraining experiments. In these settings, momentum with principal spectral regularization matches or surpasses Adam while requiring substantially less compute than full orthogonalization.

**Strengths:**

The paper tackles an important question with strong motivation. If gradients have a spiked-head, heavy-tail spectrum, then shrinking only the dominant directions is a natural way to improve stability without over-regularizing everything.

Adam, Muon, and momentum are framed as choices about how much to flatten the update spectrum. Adam and Muon make updates more uniform across directions. Because LLM gradients and momenta show a spiked-head, heavy-tail spectrum, the idea is to regularize the spike and leave the tail. That balances exploration of rare directions with stability without paying the full cost of orthogonalizing the whole space. The proposed principal spectral regularization selects a small set of dominant directions and shrinks them. Conceptually, it keeps the informative tail intact while curbing outsized directions, which the paper argues is the sweet spot for LLM pretraining. It sits between “penalize only the very top” and “orthogonalize everything,” and in practice aims to capture much of Muon’s benefit at far lower cost.

**Weaknesses:**

While the paper is interesting and offers useful insights, it has notable weaknesses. The positioning within the broader optimization literature is unclear at several points, and connections to related work and established insights are often underdeveloped.

## Novelty
### Spectral Distributions
Heavy-tailed spectral distributions in deep networks are well documented. Multiple works report spiked-head, heavy-tail structure for gradients and Hessian surrogates, e.g., [1]. Other papers motivate low-rank or structured preconditioning consistent with that view, including [2], which motivates SOAP and shows that Shampoo’s preconditioner is effectively low rank. There is also theory showing that using leading eigenvectors can improve convergence when top eigenvalues dominate compared to unpreconditioned gradient steps [3].


### The claim that partial head shrinkage can beat both unregularized momentum and full uniformization is incremental and expected

Given this background, the claim that suppressing only the principal directions reads as incremental rather than surprising.

Many popular optimizers can be viewed as preconditioned gradient methods that reduce spectral anisotropy, e.g. written as $P^{-1}g$ where $P$ approximates the Hessian. In the eigenbasis of $P$
high‑curvature directions are down‑weighted and low‑curvature directions are up‑weighted, which reduces anisotropy and pushes updates toward a more uniform spectrum.  This includes methods like K‑FAC, Shampoo, SOAP, Muon etc. Full‑matrix preconditioners explicitly rescale along all directions and approach a whitened or near‑uniform spectrum in their working basis.  Assuming heavy-tail distribution, the idea to tamp only the largest spectral components of momentum and keep the tail largely intact is an unsurprising middle ground between unregularized momentum and fully uniform updates.

So the high‑level claim — that partial head shrinkage can beat unregularized momentum and sometimes beat full uniformization — reads as an incremental twist on standard preconditioning rather than a surprising new phenomenon. The novelty is mainly in the specific, cheaper mechanism they propose for selecting and shrinking principal directions.

## Limited theory about when head‑only shrinkage should win

The central claim that shrinking only the dominant directions improves optimization lacks formal support. The paper walks a line between two motivations for PSR, one about compute efficiency and one about optimization behavior. On compute, the method replaces Newton–Schulz which in theory cuts extra FLOPs. On optimization, the method aims to "avoid amplifying noise from full uniformization" by shrinking only the spiked head while keeping the heavy tail largely intact. I agree with the first motivation because the complexity accounting is explicit, but I am not convinced by the second motivation as a general rule. The importance of small eigen directions is an open question, since those directions may correspond to hard to learn skills that still move the loss, for example rare token handling, long range interactions, niche syntactic transforms, or calibration corrections. Second order methods purposely lift such directions by whitening or preconditioning, which increases their relative step size and can help learn these skills, while the proposed method leaves the tail unnormalized and could under weight these signals when they are not noise. The evidence offered suggests the mechanism but does not isolate it or directly measure noise amplification across regimes, which matters because this mechanism is exactly what differentiates PSR from second order approaches.



## Heavy reliance on a toy problem to motivate the choice.

The Styblinski–Tang study is useful for illustrating the mechanism, but it is weak as a generalization argument. Section 3.2 constructs a power-law weighting over coordinates and adds Gaussian noise each step to emulate a spiked head with a noisy tail. In that setting, fully uniform updates will also uniformize the noise, while partially shrinking the head improves the signal-to-noise ratio. The observed sweet spot in (K) is exactly what a bias–variance tradeoff in an anisotropic, noisy problem would predict. This clarifies the intuition, but it is not evidence that the same tradeoff holds across deep networks and training regimes. A brief discussion of external validity and why the toy’s assumptions map to real LLM training would strengthen the paper.





## Typos
The paper contains many notational inconsistencies and typos, which make it harder to read. For example, “Moun” instead of “Muon”, dimension symbols and errors in lines 86, 165, 284 and many more.


[1] Zhao, Jiawei, et al. "Galore: Memory-efficient llm training by gradient low-rank projection." arXiv preprint arXiv:2403.03507 (2024).
[2] Morwani, Depen, et al. "A New Perspective on Shampoo's Preconditioner." arXiv preprint arXiv:2406.17748 (2024).

[3] Doikov, Nikita, Sebastian U. Stich, and Martin Jaggi. "Spectral preconditioning for gradient methods on graded non-convex functions." arXiv preprint arXiv:2402.04843 (2024).

**Questions:**

Please address my concerns above.

The paper contrasts PSR primarily with Muon’s Newton–Schulz orthogonalization, but the paper would be stronger if it also compared against SOAP. Because the claim centers on outperforming full uniformization at lower cost, including baselines that operationalize “target the principal directions” would sharpen the argument. SOAP is especially relevant: it explicitly leverages an approximate eigenbasis and concentrates computation on the leading directions, aiming to capture most of the benefit without fully uniformizing the spectrum.

---

> ### Author Response · Authors · 2025-11-21
>
> We thank the reviewer for the thoughtful and constructive feedback. We are grateful for this opportunity, which enables us to clarify and discuss the critical questions pointed out by the reviewers, improve our paper quality, and potentially contribute to the field. We have updated our manuscript with improved writing details and additional results, continuing to supply during the rebuttal period (due to limited computational resources and the length of pretraining experiments). Responses to the weaknesses raised by the reviewer are discussed below.
>
> **W1.1. Novelty / Spectral Distributions**: We agree with the reviewer that the heavy-tailed spectral distributions are well documented across different architectures. Nevertheless, visualizing the spectral distributions of gradient, momentum, and update matrices by AdamW and Muon not only served as a comparison less touched on in previous studies, but also demonstrated the geometric regularization perspective in designing preconditioners. This visualization highlighted a critical gap between AdamW and Muon, L-2 and L-N geometric normalization, which motivated the proposal of partial momentum orthogonalization. The contribution of this study is the implementation of PSR that bridges this gap and discusses the design choices and practical effects in this new paradigm.
>
> **W1.2. Novelty / The partial head shrinkage claim is incremental and expected**: Quantifying novelty and contributions is somehow difficult and often tricky in the existing machine learning community. Despite this, we would like to stand on our perspective, stating that most previous works focused on spectral norm (often defined as the largest eigenvalue) regularization, low-rank second-order approximations (e.g., GaLore, SOAP, and Sophia), or full-matrix spectral normalization (e.g., Muon). The concept and implementation of partial matrix orthogonalization and head shrinkage in the original gradient/momentum geometric space are an addition to the field, prompting both understanding and exploration of new mechanisms. As conventional machine learning intuitions often fall short in advanced neural architectures and large-scale deep learning practices (e.g., bias-variance trade-off vs. scaling law), we believe the observation and validation of intuitive conclusions remains a meaningful contribution to the field. We hope that the demonstration and comparison of AdamW, PSR, and Muon from a continuous geometric regularization perspective enhances understanding in large model optimization and inspires next-generation optimizers that may achieve more optimal computational trade-offs than AdamW and Muon.
>
> **W2. Limited theory about when head‑only shrinkage should win**: While theoretical studies into identifying the regimes where AdamW, Muon, or PSR should win in cutting-edge neural architectures/datasets are beyond our capabilities, we conclude from our experiments that head-only shrinkage has a clear advantage in the initial warm-up stage, and is likely to fall behind Muon in the over-training stage. PSR punished the dominant directions while maintaining the original spectral distribution, for which the convergence speed surpasses both AdamW and Muon in the early stage of training. We hypothesize that this particular setting stabilizes the update without distorting the gradient geometry too much or over-regularization.
>
> However, in the final training stage, Muon's full matrix orthogonalization uniformly amplifies and stabilizes the low-magnitude update directions (in the heavy-tail distribution of spectral distributions) that PSR strengthened relatively less. By preserving updates in these weak directions, Muon appears better equipped to navigate and settle into lower-loss regions of the landscape. Nevertheless, we argue that the lower computational cost (FLOPs and memory) of partial orthogonalization on training large-scale generative models preserves its potential for future exploration and deepening the communities' understanding of spectral preconditioning/regularization methods.
>
> **Discussions at W3/W4/Q1 are continued in the next comment.**

---

> ### Author Response · Authors · 2025-11-21
>
> **W3. Heavy reliance on a toy problem to motivate the choice**: We agree with the reviewer that the connection between the Styblinski-Tang function optimization and LLM pretraining is weak in this paper. Despite this, the power-law weight distribution settings are inherited from studies of LLM Hessian spectra (including the papers the reviewer discussed in spectral distributions), and the Gaussian noise settings are commonly used techniques in mirroring optimization problems under real-world datasets/settings. This paper centers on the geometric perspective of spectral distribution and the corresponding different regularization/preconditioning settings, re-examining them and comparing them. Thus, while the novelty and contribution of the LLM spectral distributions and ST function optimization remain vague, they served as effective tools in understanding the targeted problem in LLM optimizers. We would also appreciate the reviewer if any suggestions on conducting theoretical studies on both toy problems and LLM optimization to further strengthen our study.
>
> **W4. Typos**: The notational inconsistencies and typos pointed out are checked and fixed in the updated manuscripts. We sincerely thank the reviewer for assisting us in improving the demonstration of our study.
>
> **Q1. Comparison against SOAP**: The SOAP experiment will be supplied during the rebuttal stage, along with a discussion of the different approaches in reducing the operating dimensionality of the preconditioner, as well as other optimizers, if our computational resources satisfy the time and cost constraints.
>
> We would like to sincerely thank the reviewer for their patience and professionalism demonstrated in the review process. We are looking forward to further constructive discussions on this study and the optimization / LLM pretraining paradigm.

---

> > ### Comment · Reviewer_5xga · 2025-11-22
> >
> > > We have updated our manuscript with improved writing details and additional results, continuing to supply during the rebuttal period (due to limited computational resources and the length of pretraining experiments).
> >
> > Great. It would be much appreciated if, when you want me to re‑read revised sections, you could point me directly to the new parts.
> >
> > **1. Two motivations (compute vs optimization)**
> >
> > As I wrote in my review, the paper seems to walk a line between two motivations for PSR:
> >
> > (i) a compute efficiency motivation, where PSR is meant as a cheaper alternative to Muon’s Newton–Schulz step, and
> >
> > (ii) an optimization motivation, where shrinking only the head is supposed to “avoid amplifying noise” in the tail while still learning useful low‑energy directions.
> >
> > In your response you now seem to emphasize the computational tradeoff (in the sense that “Muon wins later” while PSR is cheaper). I am left unsure whether you are still claiming a substantive optimization advantage for head‑only shrinkage, beyond being a cheaper approximation to Muon.
> >
> > Could you clarify which of these two (or both) you regard as the main claim of the paper, and whether you still believe that keeping the heavy tail unnormalized is preferable from an optimization point of view if the goal is the strongest possible LLM rather than just a cheaper one?
> >
> > **2. The usefulness Small eigen directions**
> >
> > also want to restate a concern that remains unresolved for me. One reason I pushed on the optimization story is that small‑eigen directions are not obviously “just noise.” Methods like Muon, SOAP and the provably optimal Newton's method explicitly lift these directions by flattening the spectrum or running in a preconditioned eigenbasis, and your own experiments show that Muon ultimately achieves lower final loss, which is consistent with that story.
> >
> > If the goal is to train the best possible LLM rather than just a cheaper one, this seems like an argument for more aggressive tail lifting, not against it. I would appreciate a clearer discussion of how you see PSR’s design choice to keep the heavy tail largely intact affecting the eventual acquisition of such hard‑to‑learn skills.
> >
> > **3. Conceptual position relative to existing spectral preconditioners**
> >
> > More generally, I would expect either a clearer conceptual argument or some indication of how PSR challenges or extends our current understanding of curvature‑aware optimization. Right now that part feels under‑specified.
> >
> > My question is at the conceptual level. From a preconditioning viewpoint, PSR is still a spectral operator that takes a matrix associated with the current iterate (here the momentum), identifies its principal directions, and reweights them. In that sense it lives on the same spectrum‑flattening continuum as Shampoo, SOAP, Muon, and related methods, rather than defining a fundamentally new optimization principle. This is why I described the high‑level claim as “incremental and expected,” even though the particular implementation is novel.
> >
> > I would encourage you to be more precise in the paper about what “new paradigm” means here. To me PSR looks like a specific low‑rank spectral preconditioner operating directly on the momentum, rather than a qualitatively new optimization paradigm. If you want to claim a new paradigm, it would be important to explain how it differs in principle from existing spectral preconditioning methods, not only in implementation details.
> >
> > **4. Role of the toy problem**
> >
> > That section is a nice illustration of a bias–variance trade‑off in an anisotropic noisy problem and clearly speaks to the optimization motivation, not to the FLOP and memory story.

---

> > > ### Author Response · Authors · 2025-11-25
> > >
> > > Before attempting to resolve the reviewer's concerns, we would like to kindly remind you that we have further updated the manuscript with the ablation study results in the Table.9, Page 15 in the Appendix.D, to present and discuss the influence of the regularization factor $\eta$ and the rank proportion coefficient $m/r$, proportion of principal components computed and penalized.
> > >
> > > **1. Compute vs. Optimization**: We would like to emphasize that we are not claiming PSR to be a state-of-the-art optimization method surpassing Muon (or other optimizers like SOAP).  in computational efficiency or performance, as the empirical scope of validation is limited. In this study, we raised a hypothesis that full matrix orthogonalization could be unnecessary and implemented the PSR method to validate the hypothesis. The major claim and purpose of this paper are both (i) highlighting the potential of partial orthogonalization methods, which may function as cheaper alternatives to Muon's Newton-Schulz function, and (ii) understanding the optimization trade-offs behind head-only shrinkage / complete normalization.
> > >
> > > **2. Usefulness of small-eigen directions**: We agree with the reviewer that, based on the illustration of our study, flattening the spectrum in the heavy-tail contributes to the ultimate better performance on LLMs. Nevertheless, our proposed PSR method also demonstrates that this performance gain comes with additional computational cost, where we achieve similar performance boosting to SGD-M with only regularizing a small proportion of the spectral components. As discussed above, our study revisited the critical design choices between spectral optimization methods and revealed the different roles played by the head and tail components in the momentum spectra, using PSR as a scalable empirical probe.
> > >
> > > Furthermore, although more aggressive tail lifting may improve the ultimate LLM performance, we observe that Muon converges the slowest during the warm-up stage, whereas PSR is often the fastest. This suggests the potential value of developing adaptive spectral preconditioning methods that adjust to the evolving loss landscape, without sacrificing performance in later stages. We are willing to know if you believe the above discussion should be touched more in the main paper's discussion sections, or any other suggestions to improve the arguments of this paper.
> > >
> > > **3. Conceptual position relative to existing spectral preconditioners**: We are happy to discuss the conceptual position of PSR and why we believe it differs from previous low-rank spectral preconditioning methods. Low-rank methods are widely used for approximating high-dimensional gradients, momentum, or second-order preconditioners in the field of optimization, including GaLore, Shamoo, SOAP, etc. These methods often compute and maintain a low-rank subspace and project future computations into that subspace, reducing the computational cost. Therefore, the optimization dynamics often happen in an approximated geometry of the full parameter space. In comparison, PSR identifies the principal spectral components in the original geometric parameter space, without maintaining any low-rank projectors or preconditioners across update steps/iterations. This spectrum-flattening method not only differs principally, but also benefits the empirical analysis of the different spectral components discussed above, enabling an in-depth geometric inspection of the success and trade-offs behind Muon.
> > >
> > > **4. ** Role of the toy problem**: We sincerely thank the reviewer for agreeing with our motivation for presenting the toy problem optimization in this study, and we hope it provides an intriguing perspective for understanding and designing the spectral preconditioning method.

---

> > > > ### Comment · Reviewer_yyW5 · 2025-11-25
> > > > **Concern on the comparison against baseline**
> > > >
> > > > I also have very similar concern/confusion as Reviewer 5xga.
> > > > If there is a trade off between PSR and baseline Muon in terms of speed and final perplexity. There needs to be a cleaner study on the Pareto Frontier (a curve with FLOPs/communication/wall-clock as x-axis and Perplexity as the y-axis) across different scales trained to chinchilla optimal, maybe also with different distribution training strategy (e.g. FSDP and Tensor Parallel can have very different footprints). Otherwise, the message is very mixed and hard to judge.

---

### Official Review · Reviewer_q6g2 · 2025-10-31

**Soundness:** 3
**Presentation:** 3
**Contribution:** 2
**Rating:** 4
**Confidence:** 4

**Summary:**

This paper proposes principal spectral regularization (PSR), where only a few dominant singular vectors and their corresponding eigenvalues are penalized. This can be viewed as a partial whitening operation compared to the full whitening from Muon. The author motivate this partial operation by optimizing a toy function and vary the whitening ratio to see the effect to convergence, and discovery that by only whitening a few dominant ones can maintain most of the gains from Muon. Based on this observation, the author proposes a scheme to efficiently perform partial whitening, and evaluate this approach using LLaMA and C4 dataset.

**Strengths:**

The presentation of this paper is clear and intuitive. The proposed method seems to be valid, and the motivation is clear. Since the proposed approach aims to reduce the computational cost of Muon (is it really that expensive??), this may be of interests to practitioners with less compute power.

**Weaknesses:**

One concern I had is the contribution of this work is kind of incremental. The partial whitening of dominant eigenvectors is not too hard to think of, since one of the intuition of Muon is that it treats each components of momentum equally, and thus may overcome some bad minima during the optimization. So it is not hard to notice that instead of boosting all singular values, one can also reduce the maginitude of the top singular ones, after normalization, this should have similar effect as Muon. The proposed partial whitening approach is also not completely novel, which is a variant of classic block subspace iteration algorithm to find low-rank top eigenvectors. Despite that, it is still valuable to see this is confirmed, but not sure if this is significant enough for a publication.

Second, is NS step of Muon that expensive? Kimi utilizes Muon to train 1T model, meaning that under the distributed setup, the computational cost of Muon is manageable. This is because one cannot just use the FLOPS to determine how costy it is in practice, since NS relies on matrix multiplication, which is highly optimised and potentially be distributed a well. But from some operations, like matrix decomposition (e.g. QR), despite it may have lower FLOPS for smaller matrix, I wonder is it possible to write a customized kernel that is as efficient as matrix multiplication? Current QR uses CuSolver, which is based on efficient HouseHolder reflector methods. And QR decomposition is not as easily distributed as NS, despite that distributed QR algorithm exists. Can you offer a wall-clock time comparison, for huge model/matrix?

**Questions:**

1. For empirical evaluations, it would be interesting to see the over-train setup (36B with 1.3B model is not overtrain, since it is only 1.4x compute optimal). Maybe 1.3B with over 2-3x compute optimal?

2. Wall-clock time comparison with NS.

3. From my past experience, LLaMA with C4 dataset can be strange sometimes, i.e. some tricks is useful for this setup but fails on the other. Can you also do a similar test with 1.3B NanoGPT, openwebtext dataset and similar context length and batch size?

---

> ### Author Response · Authors · 2025-11-21
>
> We thank the reviewer for the thoughtful and constructive feedback. We are grateful for this opportunity, which enables us to clarify and discuss the critical questions pointed out by the reviewers, improve our paper quality, and potentially contribute to the field. We have updated our manuscript with improved writing details and additional results, continuing to supply during the rebuttal period (due to limited computational resources and the length of pretraining experiments). Response to each weakness and question is listed below.
>
> **W1. The contribution of this work is kind of incremental**: Thank you for raising this meaningful discussion. While whitening dominant eigenvectors is not too hard to think of, we argue that the concept and paradigm of ``partial orthogonalization'' lies outside the regime of L-2 (Adam) or L-N (Muon) geometry regularization, which may strike a balance between computational trade-offs on large-scale generative models (or other circumstances). We believe that the idea of penalizing only a proportion of spectral update directions is worthwhile to explore for the machine learning community, including the development of more principled mathematical frameworks and code implementations. The contribution of our study is to re-examine the basic concepts and highlight a potential direction for future studies.
>
> **W2./Q2. Wall-clock time comparison**: We have added Tab.3 on page 8 in the updated manuscript, demonstrating that even under a naive PyTorch implementation, the gains in runtime and memory on large-scale LLaMA layers are significant, representing a promising direction for designing new optimizers. However, the runtime on smaller models is indeed more time-consuming due to the optimization benefits of matrix multiplication. We are looking forward to future refinements, including kernel attempts and community suggestions to improve the algorithm.
>
> **Q1. It would be interesting to see the over-train setup (1.3B with over 2-3x compute, 50B-70B tokens)**: Thank you for your kind suggestions. However, our computational resources are limited, making it infeasible for us to run additional or more extensively trained experiments. The LLaMA-1.3B experiment results offer a demonstration of large-scale scaling-law–consistent training, in which the advantage of SGD-M-PSR is clear in the initial converging speed, but falls behind Muon in the final steady stage. Although we are unable to examine the properties of PSR more thoroughly due to resource constraints, we hope the existing experiment results with additional ablation studies to be supplied soon would be sufficient to demonstrate the basic properties of PSR and this new paradigm.
>
> **Q3. Can you do a similar test with 1.3B NanoGPT and OpenWebText?**: Your suggestion is greatly appreciated. We are managing to run the NanoGPT baseline experiment during this rebuttal period and are looking forward to presenting the additional results.
>
> We would like to sincerely thank the reviewer for their patience and professionalism demonstrated in the review process. We wish to emphasize that the contribution of our study lies in re-examining the optimality and computational trade-offs of the full-orthogonalization paradigm exemplified by Muon, especially in large-scale generative models. The identification, implementation, and validation of the ``partial orthogonalization'' paradigm could be considered as a valuable addition to the field, complementing the existing understandings and encouraging the exploration of more diverging methodologies.

---

### Official Review · Reviewer_5gZ8 · 2025-10-31

**Soundness:** 2
**Presentation:** 2
**Contribution:** 3
**Rating:** 2
**Confidence:** 3

**Summary:**

The paper observes that momentum updates exhibit a spiked-head-long-tail spectral distribution, meaning only a few dominant singular directions strongly influence parameter updates. Based on this observation, the paper proposes Principal Spectral Regularization (PSR), which penalizes only the top-K dominant spectral components using a lightweight block Lanczos method, instead of performing full momentum orthogonalization as in Muon. Experiments show that PSR enables SGD with momentum to outperform AdamW and match Muon’s performance while requiring less than 2% of Muon’s computation cost. The contribution lies in demonstrating that full orthogonalization is unnecessary and that selectively regularizing dominant spectral directions yields better efficiency–performance trade-offs.

**Strengths:**

1. Spectral analyses reveal a “spiked-head + long-tail” structure in momentum, helping explain why selective regularization works and providing interpretability.
2. Instead of fully orthogonalizing momentum like Muon, the paper regularizes only the dominant singular directions, greatly improving efficiency with <2% of Muon’s compute cost.
3. The method is tested both on a mathematical function optimization and on real LLM pretraining, demonstrating both theoretical soundness and practical effectiveness.

**Weaknesses:**

0. Writing: The manuscript has several presentation issues that should be addressed before publication. For instance:

    A. Multiple typos of the method name: PSR is incorrectly written as PRS in several places (e.g., Line 374/377 and Appendix).

    B. Line 165, the reference to figure is not well posted in Latex.

    C. Line 281, Algorithm 1, the mentioned function BlockBiDiagnal is not defined. Does it refer to the defined function BiDiagnal?

    D. Section 3.2, the definition of d is inconsistent and confusing. Sometimes it refers to the dimension in Styblinski-Tang function (overlapping with n), as in Line 222 and the “d=1024” in the caption of Table 1. Sometimes it refers to the number of penalized dimension (overlapping with K) as in the first line of Table 1 and in the end of Table 1’s caption.

    E. The caption of Table 3, “consistently” is misleading because AdamW performs better in some cases. Best/second-best highlighting is recommended for readability.

1. The results in Table 2 are based on a single random matrix A. Running multiple trials and reporting the average with standard deviation would increase statistical reliability.
2. The downstream verification of models is only based on LLaMA-1.3B. Evaluating additional model scales would strengthen the generality of the claim.
3. Including a wall-clock time comparison would further substantiate the claim regarding PSR’s efficiency. In addition, Muon has a FLOP overhead less than 1% compared to the forward-backward phase, which implies that its computational overhead may not be the main bottleneck. Since PSR shows noticeably lower performance than Muon and the improvement in training efficiency is not clearly demonstrated, the practical impact of PSR may appear less compelling.
4. Whether PSR is threshold-based or proportion-based (and the corresponding proportion) for LLM pretraining is not reported. Without this, results cannot be reproduced.

**Questions:**

1. Issues mentioned in C and D of Weakness 0 need clarification.

2. In Figure 3(c), some curves appear segmented in the initial stage, while others are smooth. This seems to be caused by reporting the early-stage results at a coarser interval. Could the authors clarify why the initial phase is plotted with a larger reporting interval and whether this affects the interpretation of convergence behavior?

---

> ### Author Response · Authors · 2025-11-21
>
> We thank the reviewer for the thoughtful and constructive feedback. We are grateful for this opportunity, which enables us to clarify and discuss the critical questions pointed out by the reviewers, improve our paper quality, and potentially contribute to the field. We have updated our manuscript with improved writing details and additional results, continuing to supply during the rebuttal period (due to limited computational resources and the length of pretraining experiments). Response to each weakness and question is listed below.
>
> **W0. The manuscript has several presentation issues**: The presentation issues and typo errors are corrected and updated in the new paper version. The reviewer's instructions are clear and greatly appreciated.
>
> C. On Line 281, Algorithm 1, the mentioned function BlockBiDiagonal does refer to the defined function BiDiagonal, which is now corrected.
>
> D. Regarding the ST Function, we now use $n$ to represent the function dimension and $d$ for the number of principal/penalized directions used in our toy experiment.
>
> **W1. The results in Table 2 are based on a single random matrix A**: We have added a section to the Appendix with repeated experiment results under 1000 random independent initialization runs and different weight distribution and noise settings. The standard deviation of final losses is also reported. While the statistical significance of repeated experiments is not relatively weak, the experiment served as an effective tool in understanding the problem settings and the geometric perspective of spectral distributions. The power-law weight distribution setting is inherited from studies of LLM Hessian spectra, and the Gaussian noise setting is a commonly used technique in mirroring optimization problems under real-world datasets/settings. We would also appreciate the reviewer if any suggestions on conducting further theoretical studies to strengthen our study.
>
> **W2. The downstream verification of models is only based on LLaMA-1.3B**: We have appended the downstream evaluation of LLaDA-3B and LLaDA-7B models to the Appendix.E in Tab.8. While the pretrained models in the early training stage could be less representative, the relative ranking of optimizers remains consistent with the results on LLaMA-1.3B in Tab.4: Muon performs best, followed by SGD-M-PSR, and then AdamW in compressive evaluation.
>
> **W3. Including a wall-clock time comparison**: We have added Tab.3 on page 8 in the updated manuscript, demonstrating that even under a naive PyTorch implementation, the gains in runtime and memory on large-scale LLaMA layers are significant, representing a promising direction for designing new optimizers. However, the runtime on smaller models is indeed more time-consuming due to the optimization benefits of matrix multiplication.
>
> **W4. Is PSR threshold-based or proportion-based?**: The proposed PSR regularizer in LLM pretraining experiments is proportion-based, that the identified principal directions (1/16 of the layer dimension $\min(m, n)$) are identified and shrunk to $5\%$ with matrix deflation. The parameters are specified in lines 349-350, in paragraph `SGD momentum with PSR'.
>
> **Q1. Issues mentioned in C and D of Weakness 0**: We hope the confusion caused by improper writings is now clarified in response to W0. If anything in our response remains unclear, or if our answers do not sufficiently address your questions, please let us know.
>
> **Q2. Some curves appear segmented in the initial stage**: In Figure 3, the different segmentation arises from differences in the evaluation intervals used across training configurations. We extended the evaluation interval in both sets of SGD-M experiments because the computational resources were exhausted at the time.  We have updated the figure so that all methods now share a consistent, larger evaluation interval. We also wish to re-run the pair of experiments with sufficient computational resources. Despite this, we argue that the discrepancies in early-stage reporting do not affect our conclusions, and performance differences in the later/steady-state training phases are stable.
>
> We would like to sincerely thank the reviewer for their patience and professionalism demonstrated in the review process. We wish to emphasize that the contribution of our study lies in re-examining the optimality and computational trade-offs of the full-orthogonalization paradigm exemplified by Muon, especially in large-scale generative models. Beyond L-2 or L-N geometry regularization, we believe it is worthwhile for the community to explore partial orthogonalization approaches that regularize only the dominant spectral component. Future work can expand this line of inquiry by developing more principled mathematical frameworks and more robust implementation strategies.

---

### Author Response · Authors · 2025-12-04

Dear Area Chair and Reviewers of Submission 18613,

First, we would like to express our sincere gratitude for your time and effort in managing the review process, particularly given the substantial surge in submission volumes and the unexpected security incident that happened this year. We deeply appreciate the insightful comments and constructive suggestions received from all of your reviewers and have provided detailed, point-by-point responses.

**Summary of Revisions**

In response to the constructive suggestions raised by the reviewers, we have incorporated the following significant revisions into our manuscript:

* Improved the writing, including typos, symbols, and unclear definitions or statements in the manuscript.

* Expanded the reference and discussions to related works and our motivation.

* Supplied the wall-clock time and memory consumption comparison between different orthogonalization methods on representative matrices in the main paper.

* Supplied the additional experiment results in the Appendix, including 1) repeated results on ST function optimization; 2) downstream task evaluation on LLaMA-3B/7B models trained; 3) ablation studies on the hyperparameters of SGD-M-PSR; 4) experiment results on SOAP.

* Strengthened the analysis and concluding remarks on page 10 to more clearly present the advanced understanding of spectral regularization based on our empirical results.

During the rebuttal period, we have responded to every reviewer's concern one by one. It is worth noting that multiple reviewers have recognized the novelty of our intuitive discovery and implementations. However, we also believe that not all concerns have been successfully resolved, where two of the major concerns are the limited empirical scope and the conceptual position of our discovery. We sincerely apologize for not being able to cover further experiments during this rebuttal session besides the additional ones included, as the NanoGPT run did not complete, and we are unable to examine the scaling laws on the Pareto frontier. We wish to include these results in future revisions of this work if possible. Despite these limitations, we would like to reiterate and highlight the analytical strengths of our methodology based on the results currently available.

In this study, we raised a hypothesis that full matrix orthogonalization may be sub-optimal in optimization. On top of our hypothesis, we implemented a novel principal spectral regularization method for efficient high-dimensional matrix partial orthogonalization, which differs from previous low-rank projection methods principally. While our experiment results rejected the original hypothesis in the LLM pretraining scenario, it demonstrated that with a small proportion of computational overhead, partial orthogonalization enabled the momentum optimizer to bypass Adam and achieve close performance to Muon. The empirical analytics demonstrate the different characteristics in regularizing the head or tail spectral components, which are not discussed by previous studies. Furthermore, PSR not only contributes to a better understanding of the success of Muon but also highlights the potential of a different paradigm, leading towards more computationally-efficient optimizers for next-generation large-scale generative models. We believe this study could be considered as a valuable addition to the existing field of optimization, and we hope it may motivate future work among some smartest minds.

We would like to express our sincere gratitude to all the reviewers for their constructive suggestions to our studies, and all the chairs for their commitment to ensuring the smoothness and integrity of this year's review process. Your contributions are vital to the ICLR community.

Yours sincerely,

Authors of Submission 18613

---

### Meta-Review · Area_Chair_FYKY · 2025-12-25

**Summary:**

This work, inspired by the recent success of Muon, considers "penalizing" a few dominant spectral directions in the momentum (instead of all as in Muon). However, whether the proposed method is both theoretically sound and empirically significant causes some doubts among the reviewers, and why one has to only consider penalizing a few dominant spectral directions might remain partially justified in the paper.

There are some lingering concerns:

- Figure 3 and Figure 4 do not show the entire optimization progress. Also, the precise parameter choices and the stopping criteria are not immediate from the main body of the paper. This might raise the doubt on the significance of the proposed method.
- The current work also lacks providing crystal clear guidance on how to specify the hyperparameters of the proposed Algorithm 1 and sufficient implementation details on BIDIAGONAL, SVD, QR_ORTHOGONAL in the experiments as well.
- How the proposed algorithm can be provably beneficial is missing.
- Both Reviewer yyW5 and Reviewer 5xga shared a similar concern that the proposed algorithm and baseline Muon in terms of speed and perplexity are not comprehensively evaluated in the experiments. There was a suggestion to compare performance on the NanoGPT dataset, since many related works on Muon and its variants have included this dataset in their empirical evaluations. However, the authors chose not to report the performance in this benchmark.

Overall, all the reviewers unanimously give a recommendation of a reject. I encourage the authors take the constructive feedback that they received to further strengthen their work.

**Reviewer Concerns:**

There were some presentation and grammatical issues pointed out by a couple of reviewers (Reviewer 5gZ8, 5xGA), which were fixed during the rebuttal. There were also suggestions about including walk clock time report (Reviewer 5gZ8, G6W2, yyw5); for this, while the authors included some results in the updated version of their paper, a reviewer points out that QUOTE There needs to be a cleaner study on the Pareto Frontier (a curve with FLOPs/communication/wall-clock as x-axis and Perplexity as the y-axis) across different scales trained to chinchilla optimal, maybe also with different distribution training strategy (e.g. FSDP and Tensor Parallel can have very different footprints). Otherwise, the message is very mixed and hard to judge. UNQUOTE.  Hence, the concern is not fully clarified yet.

There are also other issues that are potentially outstanding, which are listed on the summary.

**Reviewer Scores:**

All reviewers unanimously recommended rejection initially. In addition, given the lingering concerns, it is unlikely that a majority of reviewers would change their views during the rebuttal if they had been able to participate fully.

In particular, Reviewer yyW5 explicitly stated that their rating would not change, as the theoretical analysis establishing provable benefits of the proposed method is missing.

---

### Decision · Program_Chairs · 2026-01-26

Reject